# Post-catalysis structures of mitochondrial complex I with ubiquinol-10 bound in the active site

Injae Chung [1,3], Caroline S. Pereira [2], John J. Wright[1], Guilherme M. Arantes [2] ✉ & Judy Hirst [1] ✉

Respiratory complex I is a multi-subunit energy-transducing membrane enzyme essential for mitochondrial and cellular energy metabolism. It couples NADH oxidation and ubiquinone-10 ($Q_{10}$) reduction to the concomitant pumping of four protons to generate the proton-motive force that powers oxidative phosphorylation. Despite recent advances in structural knowledge of complex I, many mechanistic aspects including the reactive binding poses of $Q_{10}$, how $Q_{10}$ reduction initiates the proton transfer cascade, and how protons move through the membrane domain, remain unclear. Here, we use electron cryomicroscopy to determine structures of mammalian complex I, reconstituted into phospholipid nanodiscs containing exogenous $Q_{10}$ and reduced by NADH, to global resolutions of 2.0 to 2.6 Å. Two conformations of a reduced $Q_{10}H_2$ molecule are observed, fully inserted into the Q-binding channel in the turnover-relevant closed state. By comparing the quinone species bound in oxidised and reduced complex I structures, paired with molecular dynamics simulations to investigate the charge states of key surrounding residues, we propose a series of substrate binding poses that $Q_{10}$ transits through for reduction. Our highly hydrated structures exhibit near-continuous proton-transfer connections along the length of the membrane domain, enabling comparisons between them to assist in identifying the proton-transfer control points that are essential to catalysis.

Mammalian respiratory complex I (NADH:ubiquinone oxidoreductase) is an energy-transducing membrane-bound multimeric complex that is essential for oxidative phosphorylation and metabolism in mitochondria[1,2]. It contains 14 catalytic core subunits that are conserved in all species of complex I from prokaryotes and eukaryotes, and a cohort of 31 supernumerary subunits that are important for stability, regulation, and biogenesis[3,4]. As a major entry point to the electron transport chain, complex I uses the energy from transferring two electrons from NADH to ubiquinone (typically ubiquinone-10, $Q_{10}$) to transport protons across the inner mitochondrial membrane,

contributing to the proton-motive force ($\Delta p$) that drives ATP synthesis and transport processes. Dysfunctions of complex I lead to neuro-muscular and metabolic diseases, reflecting its central roles in metabolism, and it is also implicated in ischaemia-reperfusion injury[5,6].

High-resolution structures of complex I have now been determined from multiple mammalian species[7–12], and from species across all kingdoms of life[13–25]: they have revealed the conserved structures of the core subunits and thus the framework for the energy-converting reaction. NADH is oxidised in the mitochondrial matrix by a flavin mononucleotide at the top of the hydrophilic domain, and the two

[1]MRC Mitochondrial Biology Unit, University of Cambridge, The Keith Peters Building, Cambridge Biomedical Campus, Cambridge, UK. [2]Department of Biochemistry, Instituto de Química, Universidade de São Paulo, São Paulo, SP, Brazil. [3]Present address: Arontier Co. Ltd, Seoul, Republic of Korea.
✉e-mail: garantes@iq.usp.br; jh480@cam.ac.uk

electrons then transferred along a chain of iron-sulphur (FeS) clusters to the bound quinone (we refer to $Q_{10}$ for simplicity). The $Q_{10}$ enters the complex from the membrane and moves along a long and narrow binding channel to the active site for reduction, close to 4Fe4S cluster N2, the final cluster in the FeS chain. The substantial free energy released by reduction of $Q_{10}$ by NADH is harnessed to transport four protons across the membrane at distant sites in the membrane domain. Structures have revealed that the 'E-channel' (a conserved Glu-rich network) connects the $Q_{10}$-binding channel to the 'central axis', a hydrated network of charged residues that runs across the three antiporter-like subunits in the distal membrane domain. The antiporter-like subunits each contain an ion pair at the entry of the axis to the subunit, a central Lys/His residue, and a terminal Glu/Lys residue that are considered essential for proton pumping, and, although both their number and location remain disputed, both proton uptake and exit channels have been proposed from structural data. However, the least understood aspect of the mechanism is how the free energy from the redox reaction is harnessed and transferred to the membrane domain to coordinate and couple the redox and proton transfer reactions in the highly efficient energy-converting reaction.

Most structures of complex I are of the resting states of the enzyme, which vary between species depending on the complexity of any regulatory transitions that 'open' the canonical closed resting state. The closed turnover-ready enzyme is present in preparations of the mammalian complex, where it is referred to biochemically as the 'active' resting state, and is the sole state in preparations of complex I from *Paracoccus denitrificans*[24] and *Mycobacterium smegmatis*[23]. In contrast, open states, formed in the regulatory transition to the 'deactive' mammalian enzyme that occurs during ischaemia, and in many other species to varying extent, require reactivation to return to catalysis[11,26]. Although it is generally accepted that closed states are involved in catalysis, complex I samples frozen during turnover contain a set of reduced FeS clusters[27] whereas in the structurally defined closed resting state all the cofactors are oxidised. Therefore, even the highly characterised turnover-ready state cannot, strictly speaking, be assigned to the catalytic cycle, and the resting states give only uncertain information on the catalytic intermediates. Although samples of complex I have been frozen following the addition of substrates, to attempt to capture the structures of intermediates, the results are debated[7-9,16,18,25,26]. Challenges to interpreting the data include: the persistence of regulatory states that add to sample complexity and confound reactivation with catalysis; difficulties in correlating observed structures to catalytic steps; and experimental challenges (particularly from the extreme hydrophobicity of $Q_{10}$) in ensuring that samples were actively turning over at the point they were frozen.

Here, we aim to determine how the structures of the resting states of bovine complex I change upon NADH addition in the presence of $Q_{10}$. Our approach builds on our earlier work on oxidised bovine complex I in phospholipid nanodiscs containing $Q_{10}$ (CI-NDs)[8], in which we observed $Q_{10}$ in a pre-reaction binding pose. Here, we add a large excess of NADH to our CI-NDs, so we expect that, for NDs containing the closed/active enzyme, all the $Q_{10}$ present will be reduced to ubiquinol-10 ($Q_{10}H_2$) and the enzyme will then rest in the reduced state, perhaps with $Q_{10}H_2$ bound. For NDs containing the open/deactive enzyme, addition of NADH should stimulate reactivation and closure, although hindered by the high activation energy and low temperature of grid preparation. Therefore, our experiment does not aim to capture catalytic intermediates, but to describe the biochemically defined $Q_{10}$-hungry reduced state, which stalls after turnover waiting for another $Q_{10}$ substrate. We aim to advance knowledge of $Q_{10}H_2$ binding and investigate how the structures of the Q-binding channel, E-channel and central axis change in the reduced post-catalysis resting enzyme.

## Results

### Structures of complex I in NADH-treated $Q_{10}$-containing NDs

Complex I was reconstituted into phospholipid and $Q_{10}$-containing MSP2N2-NDs as described previously[8] and in Methods. To prepare cryo-EM grids of NADH-reduced CI-NDs (NADH-CI-NDs), 10 mM NADH was mixed with 4 nM CI at 4 °C and frozen onto the grids within $35 \pm 2$ s (see Materials and Methods). The substantial excess of NADH ensured the reduced state was maintained: continuous turnover was precluded as only three $Q_{10}$ were present per CI-ND, and no enzymes were present to recycle the $Q_{10}H_2$. Data were collected on a single grid at the UK National Electron Bio-Imaging Centre using a Gatan K3 detector on a 300 keV Titan Krios microscope (Supplementary Table 1).

NADH-CI-ND particles were first separated into three major classes, representing the three well-characterised expected states: the closed/active resting state (NADH-active), open/deactive resting state (NADH-deactive), and slack state (NADH-slack). They were then further resolved by focused classification on heterogeneous density features in the Q-binding site resulting in three closed/active substates, one homogeneous open/deactive state and one slack state (Supplementary Figs. 1 and 2), with each complex tightly enclosed by two MSP2N2 helices (Fig. 1a). The resulting models (an example is shown in Fig. 1b; see also Supplementary Fig. 3) showed that the NADH-active, NADH-deactive and NADH-slack maps display all their expected respective structural hallmarks[3,8,26,28-30], demonstrating that these hallmarks are not oxidation-state dependent. No changes were observed to the cysteine ligation of FeS cluster N2 (as reported for the reduced isolated hydrophilic domain of *Thermus thermophilus* complex I[31]). The NADH-slack state, like its oxidised counterpart, contains a cholate in its Q-binding site[8]. Recently, the slack state of bovine CI was shown to depend on isolating it in dodecylmaltoside (DDM) detergent[11]. The biochemical relevance of the slack-like states observed for bovine and ovine complex I (but not for mouse or pig or any non-mammalian species) was already under debate[7,8,11,26]; we assume here that they are detergent-linked artefacts and do not discuss them further. A cholate molecule was also observed bound peripherally, between the ND5-TMH1-2 loop and NDUFB6, as observed previously in our oxidised CI-ND structures[8], and a total of 45 phospholipid-binding positions were occupied in the NADH-active and NADH-deactive structures (Supplementary Fig. 3), with five phospholipids bound specifically to the NADH-active state at the interface of NDUFA9 with ND1 and ND3.

### NADH in the NADH-binding site

All five NADH-CI-ND structures display a clear NADH density in the flavin-containing active site, with no differences detected between the structures in this region, except for in their resolutions (Fig. 1b inset and Supplementary Fig. 4). As in earlier NADH-bound structures[7,9,16,31,32], the nicotinamide moiety of NADH is juxtaposed over the central ring of the flavin isoalloxazine in a π-stacking conformation (Fig. 1b inset and Supplementary Fig. 4) understood to facilitate direct hydride transfer from nicotinamide C4 to flavin N5. The densities for the reduced flavin are essentially indistinguishable from that of oxidised flavin in the 2.3 Å-resolution reference oxidised structure (PDB-7QSM)[8] (Supplementary Fig. 4) but nearby Phe73[NDUFV1] has changed rotamer (Fig. 1b inset), Phe209[NDUFV1] has also shifted, and residues 68–72 in NDUFV1 are displaced slightly. The adenosine diphosphate (ADP) 'handle' of NADH/NAD+, which is shared with various CI flavin-site reactants and inhibitors such as APADH/APAD+, ADP-ribose, and NADH-OH, has previously been shown to be essential for tight-binding interactions within the active site, with the substituent bound to the adenine ribose and pyrophosphate determining the quality of the nucleotide as a substrate and its effectiveness as an inhibitor[33-35]. Here, Phe73[NDUFV1], Phe209[NDUFV1] and Phe81[NDUFV1] appear to stabilise the adenine moiety of NADH by π-stacking interactions, while the ribityl hydroxyls of FMN stabilise the pyrophosphate, consistent with previous structures[7,9,15,16,31]; several water molecules around the

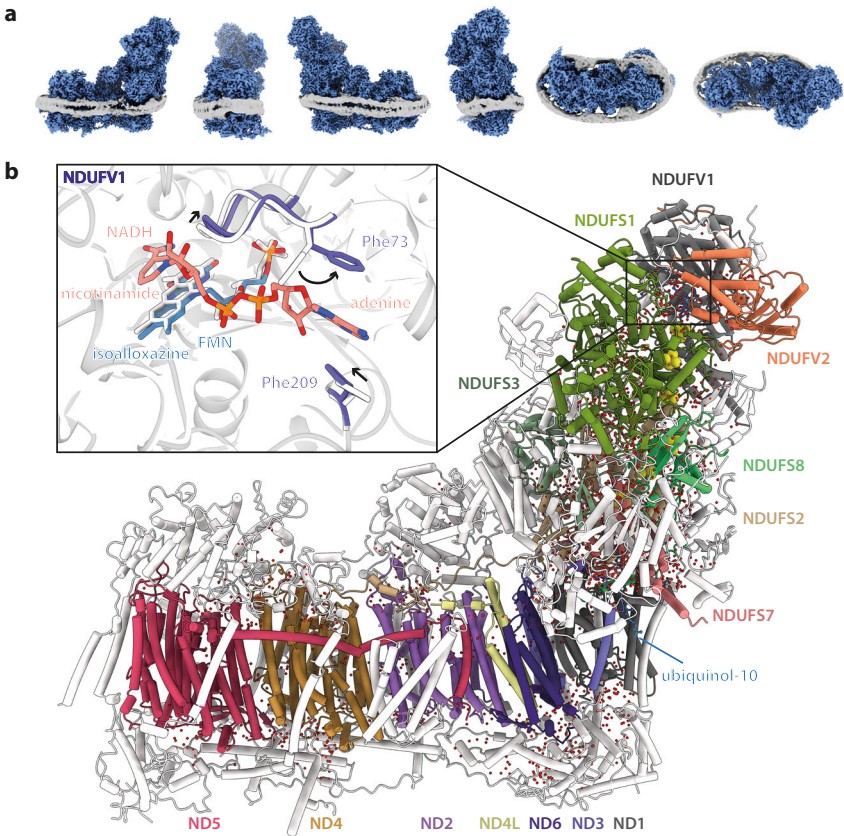

**Fig. 1 | Overview of the structure of complex I from *Bos taurus* heart mitochondria reconstituted into NDs with ubiquinone-10 and NADH. a** Three dimensional volumes of a representative complex I embedded in phospholipid NDs in six orthogonal views. Complex I and MSP2N2 helices (Gaussian filtered in UCSF ChimeraX[60]) are coloured in blue and grey, respectively. **b** The model for NADH-active-$Q_{10}$ is shown, with the 14 conserved core subunits in colour, and the 31 supernumerary subunits in white. The position of ubiquinol-10 (reduced form of ubiquinone-10) is indicated in the substrate binding site, and water molecules are shown as red spheres. Inset: an overlay of models of the oxidised active-$Q_{10}$ (PDB-7QSK)[8] and NADH-active-$Q_{10}$ CI-ND structures around the flavin site. NADH, FMN, Phe73[NDUFV1], Phe209[NDUFV1], and a stretch of residues (residues 67-73) are shown. The NADH-active-$Q_{10}$ model is in colour or grey: subunit NDUFV1 in purple, NADH in salmon, and FMN in teal. The oxidised model is in white. Arrows indicate directions of conformational change upon NADH binding. For clarity, water molecules are not shown.

nicotinamide, pyrophosphate, and pentose moieties in the oxidised nucleotide-free flavin site have been displaced by the bound NADH in the reduced site (Supplementary Fig. 4). Finally, we note that 'tight' and loose' binding poses for $NAD^+$/NADH were reported for the NADH dehydrogenase fragment (NDUFV1 and FV2) of *Aquifex aeolicus* complex I[33], with the loose position associated with a 'flipped' Ser96–Glu97[NuoF] peptide carbonyl (mammalian Glu99–Gly100[NDUFV1]) proposed to be necessary to eject $NAD^+$ and regulated by the redox state of the 2Fe-2S cluster in NDUFV2. Only the 'tight' binding pose is observed here, consistent with the much lower potential of the eukaryotic cluster[36] that maintains it in the oxidised state.

## Occupancy of the Q-site in the open/deactive state of NADH-CI-NDs

The homogenous reduced deactive state contains a well-resolved density for a DDM molecule (NADH-deactive-DDM) (Fig. 2a, d), bound within a restructured Q-binding channel that matches closely to that modelled previously for the oxidised deactive state. However, in the oxidised state a similar density was interpreted as a mixture of $Q_{10}$ and DDM, and enzyme with an apparently empty channel was also present[8]. The higher occupancy observed here suggests that DDM might have a higher affinity for the reduced than the oxidised enzyme. It forms hydrogen bonds with His55[NDUFS2] on the NDUFS2-β1–β2 loop and Arg274[ND1], a polar interaction with Glu202[ND1] on the ND1-TMH5–6 loop, and a water-mediated interaction with Glu24[ND1]; further

surrounding waters bridge it also to Gln54[NDUFS2], Tyr228[ND1] and Thr21[ND1] (Fig. 2a, d). The DDM thus interacts with multiple Q-site elements involved in the open/closed transition, and its interactions with the same restructured NDUFS2-β1–β2 and ND1-TMH5–6 loops as in the oxidised enzyme[8] suggest it stabilises the deactive state and may promote its formation and/or hinder reactivation[37]. The similar proportions of closed enzyme particles in our oxidised (17.9%) and reduced (18.6%) analyses (which were from the same mitochondrial preparation and consistent with biochemical analyses, Supplementary Fig. 1d) indicate that no substantial activation/closure occurred following NADH addition, consistent with this concept, the limited $Q_{10}$ concentration, and the low sample temperature, which is also expected to hinder reactivation. Our results reinforce the message that care must be taken when interpreting structural changes to the Q-binding site observed in the presence of DDM.

## Occupancy of the Q-site in the NADH-active enzyme

A $Q_{10}$ species is observed fully inserted into the Q-binding channel of the closed/active enzyme in two distinct conformations, NADH-active-$Q_{10}$ and NADH-active-alt$Q_{10}$ (Fig. 2a–c). Both densities extend from the tip of the channel, adjacent to the His59[NDUFS2] and Tyr108[NDUFS2] residues that are proposed to hydrogen bond to the Q headgroup and donate protons to it during catalysis[38–41], to where it meets the membrane in ND1. Due to the excess NADH and limiting $Q_{10}$, we assign both to the ubiquinol-10 ($Q_{10}H_2$) reduced form. No empty channel is detected

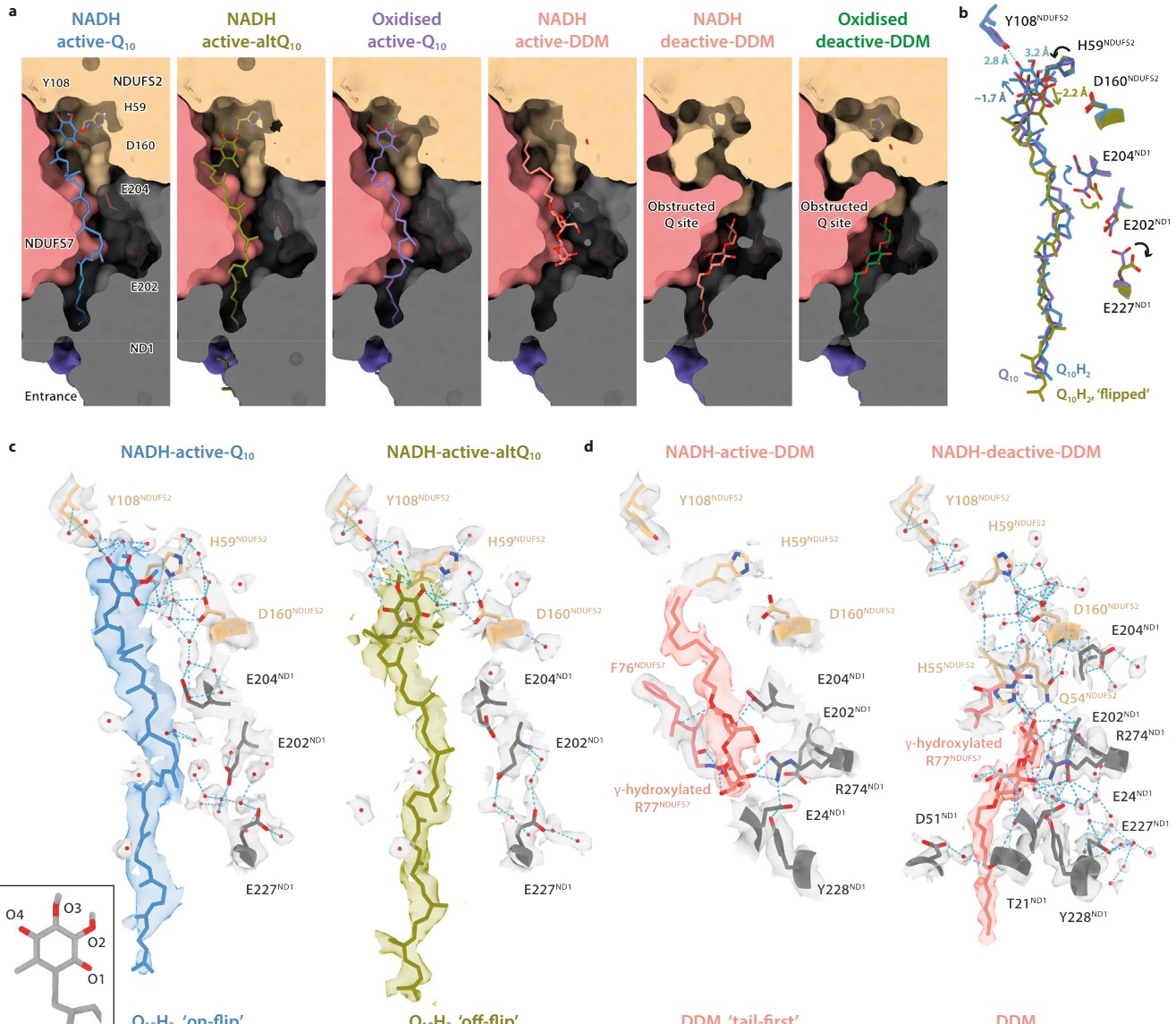

**Fig. 2 | Conformations of ubiquinol-10 and DDM bound in NADH-CI-NDs.**
**a** Clipped view of the substrate-binding site in bovine CI-NDs encapsulated by subunits NDUFS2 (beige), NDUFS7 (salmon), ND1 (grey), and ND3 (navy). Bound Q species and DDM are shown in colour as indicated; oxidised active-$Q_{10}$ is from PDB-7QSK and oxidised deactive-DDM from PDB-7QSM [8]. The site is obstructed in the deactive states by the reordered NDUFS2 β1–β2 loop. **b** Overlay of bound $Q_{10}H_2$ and $Q_{10}$ in NADH-treated and oxidised CI-NDs, respectively, aligned on subunit NDUFS2 in the same view. Models are coloured as in (**a**) and arrows denote the movement of Q-headgroups following reduction. **c, d** Coulomb potential densities attributed to (**c**) $Q_{10}H_2$, (**d**) DDM, water molecules within 5 Å, and residues including but not restricted to His59$^{NDUFS2}$, Tyr108$^{NDUFS2}$ and Asp160$^{NDUFS2}$ in NADH-CI-NDs, coloured as in (**a**). Map densities are shown at thresholds (σ) of 4 (NADH-active-$Q_{10}$) and 3 (NADH-active-alt$Q_{10}$), 4.5 (NADH-active-DDM) and 2 (NADH-deactive-DDM) in UCSF ChimeraX[60]. Hydrogen bonds (blue) were identified using the *hbonds* command in UCSF ChimeraX[60]. Figure inset (black box) indicates the labelling scheme for oxygen atoms.

here, suggesting higher occupancy for $Q_{10}H_2$ in the reduced enzyme than observed previously for $Q_{10}$ in the oxidised enzyme (where classification of the closed population attributed ~60% of the particles to the empty class[8]). No conformational changes are observed in the NDUFS2-β1–β2 loop that carries His59$^{NDUFS2}$, which was observed to move either towards (PDB-6ZKC)[7] or away (PDB-7V2E)[9] from Asp160$^{NDUFS2}$ upon addition of substrates to the closed/active states of ovine and porcine complex I.

In the NADH-active-$Q_{10}$ pose, the $Q_{10}H_2$ headgroup is flipped-over and shifted ~1.7 Å toward Tyr108$^{NDUFS2}$ relative to $Q_{10}$ in the oxidised active-$Q_{10}$ structure (PDB-7QSK)[8] (Fig. 2b). $Q_{10}H_2$-O4 (see Fig. 2 inset for nomenclature) is in a hydrogen bonding conformation with the Tyr108$^{NDUFS2}$-O (~2.8 Å, the sidechain hydroxyl-O) and an adjacent water molecule (~3.0 Å). However, His59$^{NDUFS2}$ is not within hydrogen bonding distance of either of the $Q_{10}H_2$ hydroxyl oxygens (≥ ~4.5 Å)

(Fig. 2b, c) but rather rotated by ~90° so that its imidazole ring lies parallel to the $Q_{10}H_2$ headgroup. The imidazole-Nδ is positioned ~3.9 Å away from Asp160$^{NDUFS2}$ (in comparison to ~5.0 Å in oxidised active-$Q_{10}$)[8] but the two residues are again bridged by an intermediary water (Fig. 2c). If the $Q_{10}H_2$ headgroup in NADH-active-$Q_{10}$ is flipped over to match the oxidised active-$Q_{10}$ structure[8], the two methoxy groups no longer fit well into the density, the interaction between $Q_{10}H_2$-O4 and Tyr108$^{NDUFS2}$-O is lost, and the $Q_{10}H_2$-O1 involved in a hydrogen bonding network with surrounding waters is replaced by the 5-methyl; His59$^{NDUFS2}$ remains too distant from $Q_{10}H_2$-O1 (~5.0 Å) for a viable hydrogen bond (Supplementary Fig. 3). Attempts to reposition the headgroup to create hydrogen bonds to both the Tyr and His without moving the isoprenoid chain out of its density were unsuccessful.

The $Q_{10}H_2$ headgroup in the NADH-active-alt$Q_{10}$ structure is translated ~3.9 Å down the channel and flipped-over with respect to

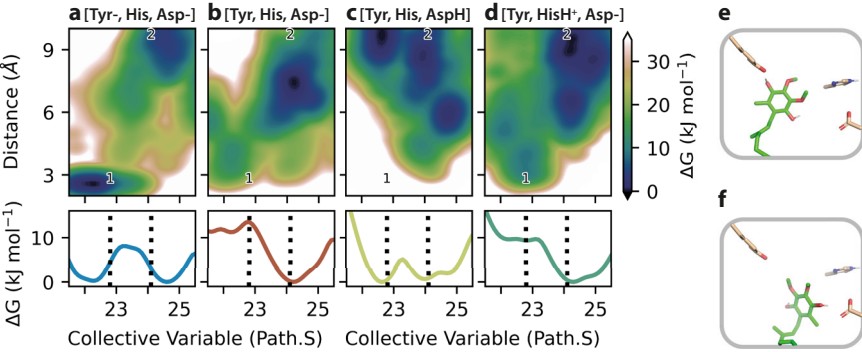

**Fig. 3 | Free energy profiles from molecular dynamics simulations.** Four combinations of sidechain protonation states, with His59[NDUFS2] neutral (His, Nε protonated) or di-protonated (HisH+, both Nδ and Nε protonated), Tyr108[NDUFS2] neutral (Tyr) or ionised to tyrosinate (Tyr-), and Asp160[NDUFS2] protonated (AspH) or ionised (Asp-), were simulated. (**a**) shows results for the [Tyr-, His, Asp-] combination (total charge -2), (**b**) for [Tyr, His, Asp-] (charge -1), (**c**) for [Tyr, His, AspH] (charge 0) and (**d**) for [Tyr, HisH+, Asp-] (charge 0). The collective variable (Path.S) describes the Q-head position along the binding channel, as illustrated for Path.S = 22.5 in (**e**) and Path.S = 24.5 in panel (**f**), both obtained from the [Tyr-, His, Asp-] state simulation. In (**a–d**), two-dimensional profiles are shown in upper panels for the distance between $Q_{10}H_2$-O4 and Tyr108[NDUFS2]-O, and one-dimensional profiles along the Path.S variable are shown in lower panels. Dashed lines at Path.S = (22.8 and 24.1) and numbers (1 and 2) correspond to the Path.S and distance observed in cryo-EM models NADH-active-$Q_{10}$ and alt$Q_{10}$, respectively.

the NADH-active-$Q_{10}$ pose (Fig. 2b, c). The headgroup density is poor but the isoprenoid density is clear, so this structure likely represents an averaged[42] mixture of $Q_{10}H_2$ binding poses. Intriguingly, a further class (NADH-active-DDM) containing a contiguous density consistent with an adventitiously bound DDM is also observed (Fig. 2a, d). In contrast to in NADH-deactive-DDM, the DDM is inserted tail-first and binds at the top of the Q-binding channel with its aliphatic tail approaching His59[NDUFS2] and Tyr108[NDUFS2]. There are no previous examples of a DDM molecule observed binding in the closed/active state of complex I, rather than in the open/deactive state, although the occupancy as observed here is low (~22%) (Supplementary Table 1 and Supplementary Fig. 1). As the DDM molecules resolved in the active and deactive states are inserted the opposite way round, they probably enter the binding channel separately, in the orientation they are observed. Most likely the DDM, which may confound interpretations of Q-site densities, has been missed in earlier structures due to its low population.

## Molecular simulations of $Q_{10}H_2$ in NADH-active states

Expanding on our structural data, MD simulations with enhanced sampling were carried out to explore the charge states of groups in the Q-binding site when cluster N2 is reduced. The same starting structure (but with N2 set into the reduced state) and approaches were used as for simulations previously carried out for the oxidised active-$Q_{10}$ model[8], which displays negligible differences in relevant side-chain positions relative to the structure reported here. As before, a pathway collective variable (CV) was used to describe the Q-head position along the binding channel (Path.S), with lower values representing a more buried Q-head, closer to Tyr108[NDUFS2]. The $Q_{10}$ molecule was assumed to always be in the $Q_{10}H_2$ (quinol) state and four charge states, with progressive protonation of the Tyr108[NDUFS2], Asp160[NDUFS2] and His59[NDUFS2] sidechains, were tested.

Figure 3 shows that only the free energy profile obtained for the [Tyr-, His, Asp-] charge state exhibits a well-defined minimum, consistent with a 2.8 Å hydrogen bond between $Q_{10}H_2$-O4 and Tyr108[NDUFS2]-O, and a stable Q-head position (Path.S = 22.3, Fig. 3a) that matches the NADH-active-$Q_{10}$ model. The other charge states do not form a stable hydrogen bond. In [Tyr, His, Asp-] (Fig. 3b) and [Tyr, HisH+, Asp-] (Fig. 3d), $Q_{10}H_2$-O4 and Tyr108[NDUFS2]-O may transiently approach, but there is no clear free energy minimum for the Q-head position. Charge state [Tyr, His, AspH] (Fig. 3c) shows a free energy minimum (Path.S = 22.6), but in this state $Q_{10}H_2$-O4 and Tyr108[NDUFS2]-O do not approach. Additional structural properties (Supplementary Fig. 5) also show better agreement between the NADH-active-$Q_{10}$ model and the [Tyr-, His, Asp-] charge state. The average hydration of

reactive groups in this charge state (Tyr coordinated to ~3 waters, His to ~3 waters, and Asp to 4–5 waters) closely matches the hydration observed in the NADH-active–$Q_{10}$ model (Fig. 2c). Thus, our simulations suggest the experimental NADH-active-$Q_{10}$ structure is in the [Tyr-, His, Asp-] charge state. The $Q_{10}$ has been reduced to $Q_{10}H_2$ by transfer of two electrons (from cluster N2) and two protons, with the Tyr and Asp as the two deprotonated sites, relative to the previously determined oxidised active-$Q_{10}$ state (which simulations indicated was in the [Tyr, His, AspH] state[8]). The $Q_{10}H_2$-O4 hydroxy is assumed to have accepted its proton from Tyr108[NDUFS2], to which it remains strongly hydrogen bonded in the tyrosinate anion (Tyr-) form (consistent with their respective $pK_a$ values in bulk solution) whereas the identity of the direct proton donor to $Q_{10}H_2$-O1 is not defined by our structure (Fig. 3e). It is usually assumed to be His59[NDUFS2], in which case a productive interaction between the $Q_{10}H_2$-O1 and one of the two His-N centres must have formed and broken before the formation of the states observed here. Alternatively, we cannot exclude that the proton is transferred by water molecules, via the hydrated network that is observed between O1, His59[NDUFS2] and Asp160[NDUFS2].

The four protonation states also show free energy minima (lower panels in Fig. 3a–d) in the interval of Path.S = 24.0 to 24.5 (Fig. 3f), compatible with the Q-head position observed in the NADH-active-alt$Q_{10}$ model. Free energy profiles for the distance between $Q_{10}H_2$-O4 and Tyr108[NDUFS2]-O (upper panels in Fig. 3a–d) and for additional structural properties (Supplementary Fig. 5) also show minima corresponding (within 5 kJ mol⁻¹) to distances observed in the alt$Q_{10}$ model for all charge states. Thus, in the simulations, the $Q_{10}H_2$-binding pose in the NADH-active-alt$Q_{10}$ state is compatible with all four charge states, and a mixture may also be present.

## Mapping of candidate proton-transfer routes in the closed/active reduced state

A 4 Å-distance cutoff was used to identify potential Grotthuss-competent connections between Asp, Glu, His, Lys, Ser, Thr, and Tyr sidechains and water molecules, and interpreted to identify candidate proton pathways for comparison with previous analyses[10,11,18,24,43]. The candidate pathways detected, which do not take interaction geometries into account, are shown in detail in Supplementary Fig. 6a, b and summarised in Table 1.

The overall connectivity in the NADH-active-$Q_{10}$ structure (Supplementary Fig. 6a) is similar to observed previously[7–9,14,16], but it is unusual to observe a complete connection from Tyr108[NDUFS2] and His59[NDUFS2] at the tip of the Q-site to the first charged residue of the antiporter-like subunit ND2 (TMH5-Glu34[ND2]) via Glu202[ND1], the E-

**Table 1 | Distance-based analyses of potential Grotthuss-competent connections formed in key elements of complex I**

| | Q | E | α | I | ND2 | | | | ND4 | | | | ND5 | | | | Matrix | | |
|---|---|---|---|---|---|---|---|---|---|---|---|---|---|---|---|---|---|---|---|
| | | | | | S | C | T | I | S | C | T | I | S | C | T | X | 2 | 4 | 5 |
| NADH-active-Q₁₀ (this study) | ● | ● | ● | ● | ○ | ● | ● | ● | ○ | ● | ● | ○ | ● | ○ | ✕ | ○ | ✕ | ○ | ○ |
| NADH-active-altQ₁₀ (this study) | ○ | ○ | ○ | ● | ○ | ● | ○ | ● | ○ | ● | ● | ○ | ● | ○ | ✕ | ○ | ✕ | ○ | ○ |
| Closed, mouse (PDB-8OM1) | ○ | ○ | ○ | ● | ● | ● | ● | ○ | ○ | ○ | ● | ● | ● | ○ | ✕ | ○ | ✕ | ● | ● |
| Closed, bovine (PDB-8Q48) | ○ | ○ | ○ | ● | ○ | ○ | ○ | ○ | ○ | ○ | ● | ● | ● | ○ | ✕ | ○ | ✕ | ○ | ○ |
| Closed, *P. denitrificans* (PDB-8QBY) | ○ | ○ | ○ | ● | ● | ○ | ● | ● | ● | ● | ● | ○ | ○ | ✕ | ● | ○ | ✕ | ○ | ○ |
| Closed, *E. coli* (PDB-7Z7S) | ○ | ○ | ○ | ● | ● | ○ | ○ | ○ | ○ | ○ | ● | ○ | ● | ✕ | ○ | ○ | ✕ | ○ | ○ |
| NADH-deactive (this study) | ○ | ● | ✕ | ● | ● | ● | ● | ● | ● | ● | ● | ● | ● | ● | ✕ | ○ | ✕ | ● | ● |
| Open, *Y. lipolytica* (PDB-7O71) | ○ | ○ | ○ | ● | ● | ● | ● | ● | ○ | ● | ● | ● | ● | ○ | ✕ | ○ | ○ | ○ | ● |

Structures were selected based on their resolution and confident modelling of substantial numbers of water molecules.
Key:
Black circles = Connected (within 4 Å distance); Open circles = Not connected but the gaps are all unobstructed, so the possibility of a connection exists; X = Not connected and at least one gap is obstructed.
Q = from Tyr108$^{NDUFS2}$/His59$^{NDUFS2}$ to Glu204$^{ND1}$ (Q-site); E = Glu204$^{ND1}$ to Asp66$^{ND3}$ (E-channel); α = Asp66$^{ND3}$ to Glu70$^{ND4L}$ (around ND6-TMH3); I = across the interface to the next subunit (ND4L to ND2, ND2 to ND4, ND4 to ND5); S = across the TMH5-Glu/TMH7-Lys salt bridge; C = from the TMH7-Lys to the central His (or Lys in ND2); T = from the central residue to the terminal TMH12-Lys or Glu; X = ND5 only, connection from the terminal residue to the IMS; 2, 4, 5 = connections from the matrix to the central residue in ND2, 4, 5.

channel, Asp66$^{ND3}$ and Glu70$^{ND4L}$. The greater connectivity may be related to the high resolution, which encourages the modelling of water molecules. In the NADH-active-altQ₁₀ model (Supplementary Fig. 6b and Table 1) the corresponding network is fragmented − but by unobstructed gaps that may simply arise from lack of resolution of dynamic water molecules or sidechains.

The ca. 200 Å−long central axis of charged residues in ND2, ND4, and ND5 is highly hydrated in both NADH-active structures, forming candidate connections across subunit interfaces, between the conserved TMH5-Glu and TMH7-Lys ion pairs, the central TMH8-Lys/His, and the TMH12-Glu/Lys residues (Supplementary Fig. 6a, b and Table 1). Again, the connectivity in NADH-active-Q₁₀ is more complete than in NADH-active-altQ₁₀. The connectivity across ND4 in both structures shows very clearly how the central axis crosses the subunit via TMH7b-His220$^{ND4}$, at the bottom of the proposed matrix-uptake pathway, rather than by TMH8-Lys237$^{ND4}$. Consistently, the H220F$^{ND4}$ mutation in *P. denitrificans* abolished CI catalysis[44], whereas the K237A$^{ND4}$ and K237Q$^{ND4}$ mutations in *E. coli* and *P. denitrificans* had only partial effects[18,44–46]. In ND5, the incomplete ND4-side connection to the central residue TMH8-His248$^{ND5}$ (spatially equivalent to TMH7b-His220$^{ND4}$) may be due to unresolved waters, but the onward connection is blocked between His248$^{ND5}$ and Thr306$^{ND5}$. His248$^{ND5}$ has been proposed to switch between two positions both in CI and in the evolutionarily-related Mrp-type antiporters[18,47,48], with the current pose directing protons from the matrix toward TMH7-Lys223$^{ND5}$, rather than toward TMH12-Lys392$^{ND5}$.

In every antiporter-like subunit, the upper section of the proposed matrix-side half-channel − enclosed by TMHs 7b, 8, 10, and 11 − is open to the matrix (Fig. 4), but no complete connections from the matrix to the central axis are observed due to a series of obstructed (ND2) or unobstructed (ND4, ND5) gaps in candidate regions (Supplementary Fig. 6a, b and Table 1). In ND2, simulations in *Y. lipolytica* CI[16] have proposed that a string of water molecules connects the central TMH8-Lys135$^{ND2}$ to TMH7b-Thr119$^{ND2}$ but here the connection is sterically obstructed by TMH8-Leu131$^{ND2}$ and TMH7b-Val115$^{ND2}$. Interestingly, Val115$^{ND2}$ corresponds to the Leu in the Leu-Trp-His gate that has been suggested to be involved in proton gating in mouse and *Y. lipolytica* complex I[10,14,49,50], and the activity of the Leu-to-Ala variant in ND4 was

low in *P. denitrificans* CI[44]. Where the proposed channel connects ND2 to the matrix, Arg176$^{ND2}$ is highly hydrated and exposed at the interface between ND2, NDUFA10, and the N-terminal loop of NDUFS2 (Supplementary Fig. 6a, b and Fig. 4d, g). In ND4, a substantial network that includes Tyr334$^{ND4}$, Glu335$^{ND4}$ and His83$^{ND4}$ (between TMHs 7b, 8, 10, and 11, just below the NDUFS2 N-terminus) connects TMH8-Ser228$^{ND4}$ (which is close but not connected to TMH7b-His220$^{ND4}$) to the matrix, extending through the supernumerary subunits (Supplementary Fig. 6a, b and Fig. 4c, f). In ND5, the equivalent network is more limited, extending from TMH8-His248$^{ND5}$ and TMH8-Ser244$^{ND5}$ to matrix-side residues TMH10-Lys299$^{ND5}$, Thr241$^{ND5}$, Tyr107$^{ND5}$ and TMH7b-Glu238$^{ND5}$ (Supplementary Fig. 6a, b and Fig. 4b, e).

### Water networks in the open/deactive reduced state

In the DDM-bound deactive state (NADH-deactive), the ubiquinone-binding site is highly hydrated, with a continuous network up to just before the start of the E-channel, where there is an unobstructed gap between the water network around Asp160$^{NDUFS2}$ and the extensive E-channel network connecting Glu204$^{ND1}$ to Asp66$^{ND3}$ (Supplementary Fig. 6c and Table 1). As observed previously in oxidised deactive CI-NDs[8], detergent-solubilised CI[7,14,16], proteoliposome-reconstituted CI[11], and simulations[41,51], the deactive-specific ND6-TMH3 π-bulge disrupts the onward connection to Glu34$^{ND4L}$ and the antiporter-like subunits. From Glu34$^{ND4L}$, the central axis is then linked across subunits ND2, ND4, and partially across ND5 to TMH8-His248$^{ND5}$ in a remarkably continuous network (Supplementary Fig. 6c and Table 1). As observed in the NADH-active states, there is no direct connection to the matrix in ND2, even though the ND2 half-channel is matrix-exposed. However, both matrix-side hydrations in the ND4 and ND5 half-channels now appear to be connected to the central axis, as well as to each other (Supplementary Fig. 6c).

### Discussion

#### Relationship of the reduced Q₁₀H₂-bound active state to the catalytic cycle

Here we observe a Q₁₀H₂ molecule fully occupying the long and amphipathic Q-binding site in closed, NADH-reduced CI. We propose the system is a 'post-reaction' state in which Q₁₀H₂ pauses after

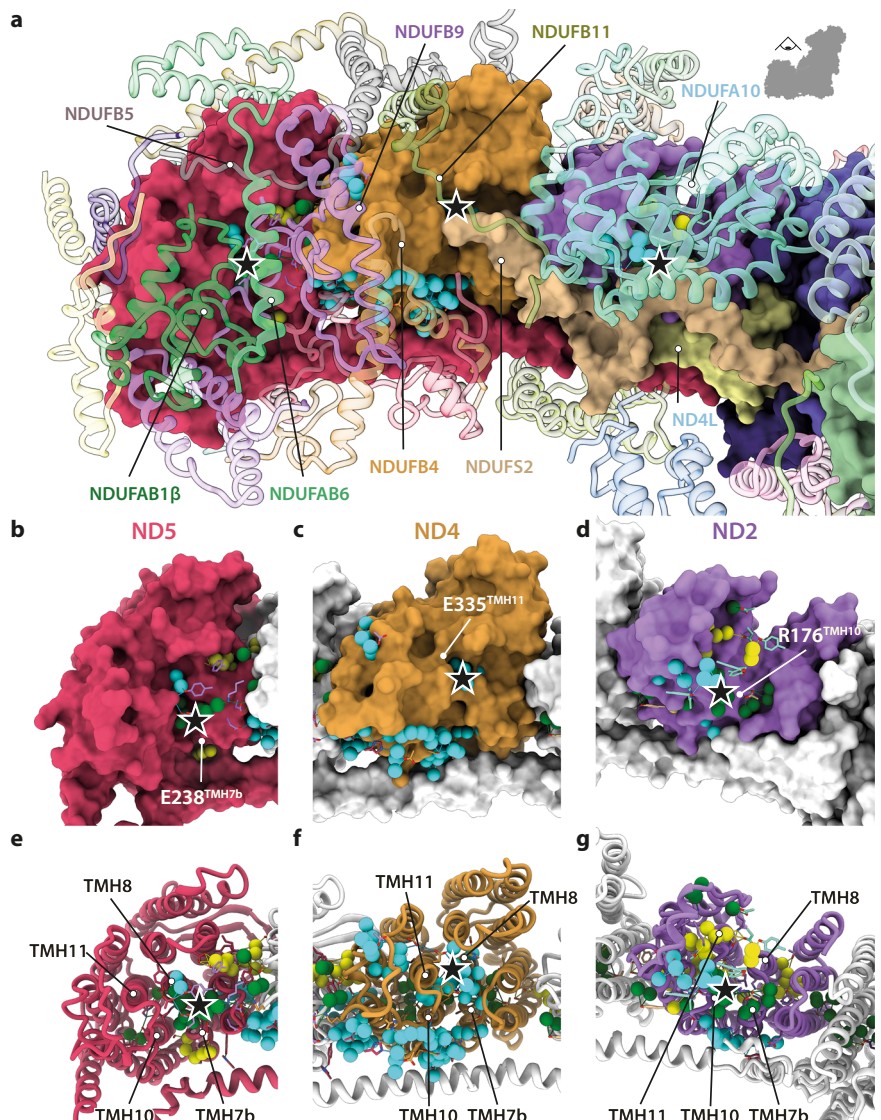

**Fig. 4 | Matrix-side 'entry points' to the antiporter-like subunits ND2, ND4, and ND5 for proton transfer. a** The membrane arm of CI (NADH-active-$Q_{10}$). Core subunits are shown as surfaces, supernumerary subunits as cartoons, and water molecules as spheres (coloured as in Supplementary Fig. 6a). **b–g** Matrix-exposed proton inlets (indicated as ★) in subunits (**b**, **e**) ND5, (**c**, **f**) ND4, and (**d**, **g**) ND2; TMHs 7b, 8, 10, and 11 – which form the proton entry cavity – are labelled. To aid site identification, one residue adjacent to the top of each matrix-side entry point is labelled in (**b–d**).

reduction, with a single hydrogen bond between $Q_{10}H_2$-O4 and Tyr108[NDUFS2]-O. In this post-reaction state, our simulations indicate the quinol is hydrogen-bonded to the Tyr108[NDUFS2] tyrosinate in the [Tyr-, His, Asp-] state. Previously, an equivalent approach indicated that the oxidised structure with $Q_{10}$ in a pre-reaction state was in the [Tyr, His, AspH] charge state, and so our conclusion is consistent with the abstraction of two protons from the Tyr and His/Asp, upon reduction by two electrons. $Q_{10}H_2$ binding in the site likely stabilises this protonation state, with its dissociation preceding or concurrent with proton uptake from the matrix to reload the site.

It is reasonable to ask why the quinol has not dissociated. One possibility is that we have trapped an intermediate that precedes product release and initiation of the proton translocation cascade. For example, it has been proposed that a quinol species moves down the site and positions its headgroup at the top of the E-channel to trigger proton translocation[16,52], and we may have trapped a state preceding this transition. It is possible that the transition is initiated by reprotonation of the Tyr P and Asp to release the quinol, which has not yet occurred. The tightly wrapped MSP2N2 helices and limited lipid phase

of the ND may have assisted in trapping the intermediate state by hindering substrate exchange or a coupled catalytic step, or by altering charge distribution across the membrane. Alternatively, it is possible that the whole catalytic reaction has completed, returning the system to the (reduced) resting state, either simply awaiting quinol dissociation, or with quinol bound 'back' into the site in a product-inhibition mode.

**Comparison and classification of Q binding poses**
Comparison of our oxidised and reduced structures of $Q_{10}$- and $Q_{10}H_2$-bound species in CI-NDs reveals that two protons are abstracted from the protein when the quinol is formed, and suggests that either quinol binding stabilises the proton-deficient state, or that catalysis has paused in a state that precedes proton uptake. However, wider comparisons with Q-binding poses in published structures (Table 2) raise some uncertainties.

It is logical to expect all closed resting states of oxidised CI to exhibit the same Q-binding pose, but this is not the case. In *P. denitrificans*, the $Q_{10}$ headgroup was observed midway down the channel[24],

**Table 2 | Experimental structures of closed CI with bound Q species**

| Model | 1 | 2 | 3 | 4 | 5 | 6 | 7 | 8 | 9 | 10 | 11 | 12 | 13 |
|---|---|---|---|---|---|---|---|---|---|---|---|---|---|
| PDB | 9SMF | 9SMG | 7QSK | 8Q48 | 8Q45 | 7V2C | 7V2R | 7V2H | 8UEP | 6ZKC | 8ESZ | 7Z7S | 7Z8O |
| Organism | Bov | Bov | Bov | Bov | Bov | Porc | Porc | Porc | Porc | Ovi | Dros | Ecoli | Ecoli |
| Q | $UQ_{10}$ | $UQ_{10}$ | $UQ_{10}$ | $UQ_{10}$ | $UQ_{10}$ | $UQ_{10}$ | $UQ_1$ | dQ | $UQ_{10}$ | dQ | $UQ_{10}$ | dQ | dQ |
| NADH | + | + | – | – | – | – | + | + | + | + | – | + | + |
| Preparation | ND | ND | ND | $PL_{out}$ | $PL_{in}$ | Det | Det | Det | In situ | Det | Det | Det | Det |
| RMSD | – | 0.03 | 0.23 | 0.27 | 0.27 | 0.26 | 0.22 | 0.21 | 0.42 | 0.48 | 0.67 | 0.91 | 0.92 |
| Path.S | 22.8 | 24.1 | 23.6 | 22.9 | 22.8 | 22.9 | 22.8 | 23.1 | 23.2 | 23.2 | 23.6 | 24.1 | 24.1 |
| Path.Z (–) | 0.20 | 0.23 | 0.23 | 0.21 | 0.20 | 0.21 | 0.21 | 0.22 | 0.23 | 0.23 | 0.23 | 0.21 | 0.21 |
| Ring-flip | On | Off | Off | On | On | On | On | Off | Off | Off | Off | – | - |
| Mode | I | K.2 | J.2 | I | I | I | I | J.1 | J.1 | J.1 | J.2 | K.1 | K.1 |

Models were obtained by various research groups using various preparations (ND, nanodiscs; PL, proteoliposomes; Det, detergent micelles) from various organisms (bovine, porcine, ovine, *Drosophila* and *E. coli*) with various Q species (ubiquinone-1, 10 and dQ, decyl-ubiquinone), with and without the presence of NADH. Root mean-squared deviations (RMSD, Å) are for main-chain Cα in NDUFS2, NDUFS7 and ND1. Path.S and Path.Z (in nm²) are collective variables that describe the internal position of the Q-head in the binding channel, used to classify structures into binding modes I, J and K (Fig. 5). See also Supplementary Table 2.

whereas in closed structures from a wide variety of preparations (Table 2) it is bound at the top of the channel[8,9,11,12,22]. The menaquinone bound in closed *M. smegmatis* CI was observed in both places[23]. In Table 2 the main binding poses for the Q-headgroup at the top of the channel are classified by its location along the channel (Path.S), tilting and turning (Path.Z), and its 'flip' state[8,40]. All the observed headgroups (except in *E. coli* CI) are essentially coplanar and easily separated into an 'on' flipped state, in which a reactive Tyr–Q hydrogen bond can form, and the Q-methyl points towards the Tyr, and the opposite 'off' flipped state. Despite the common closed state of all the structures considered, the range of binding modes adopted by Q species at the top of the channel is striking and currently lacks explanation – including the different poses detected in oxidised bovine CI between ND and PL preparations[8,11], within essentially indistinguishable protein structures. Possibilities include subtle effects in the environment, for example, the lateral pressure and phospholipid charges of the membrane in PLs, or NDs, and detergent restricting Q exchange, modulating the global protein dynamics, or changing the relative stabilities of different substates. Alternatively, the reduction state of the system may be altered by the cryo-EM electron beam[53], blurring the differences between 'oxidised' and 'reduced', or cryo-EM particle classification procedures that aim to separate similar densities from a mixture may introduce artefacts. This comparison is a caveat on our conclusions, and highlights the crucial importance of reproducing results across different systems to reveal weaknesses in understanding and stimulate new approaches.

Considering these observations, we describe three Q-headgroup binding modes at the top of the channel (Table 2 and Fig. 5). Mode I, observed here in NADH-active CI-NDs, has an on-flipped Q-headgroup close to cluster N2 and an $Q_{10}H_2$-O4 to Tyr108$^{NDUFS2}$-O hydrogen bond; it was also observed for bovine CI in PLs (oxidised, $UQ_{10}$)[10] and CI in the porcine supercomplex (oxidised, $UQ_{10}$ and reduced, $UQ_1$)[9], although with variations in the position and interactions of His59$^{NDUFS2}$. Mode J, characterized by an off-flipped Q headgroup and weak hydrogen bonds between His59$^{NDUFS2}$/Tyr108$^{NDUFS2}$ and the Q 2/3-methoxys, is exhibited by five structures in two sub-groups: J.1 was observed in ovine and porcine CI (reduced, dQ)[7,9], and in porcine CI (reduced, $UQ_{10}$) in situ[12]; and J.2 in bovine CI in NDs (oxidised, $UQ_{10}$)[8] and *Drosophila* CI (oxidised, $UQ_{10}$)[22]. Finally, in Mode K, the *E. coli* sub-group K.1 (reduced, dQ)[18] is distinct from the bovine NADH-active-alt$Q_{10}$ K.2 structure (reduced, $UQ_{10}$) that has an off-flipped headgroup. It is likely the Met-to-Val (Met60$^{NDUFS2}$ in bovine) substitution in the *E. coli* Q-channel alters the Q-tail position and Q-head bending (model 12, see Fig. 5b). Mode K has no hydrogen bonds between Tyr/His and the Q-head. In summary, Modes I, J and K exhibit increasing distances from

cluster N2 and decreasing hydrogen bond contacts with active site residues, and the Q-headgroup flips between Modes I and J.

Finally, of the 'reactive conformations' proposed previously in simulations (the dual hydrogen-bonded binding mode[39,40,54] with hydrogen bonds to the Q-carbonyls from both Tyr108$^{NDUFS2}$ and His59$^{NDUFS2}$, and single hydrogen-bonding modes with a hydrogen bond from either Tyr108$^{NDUFS2}$ to $Q_{10}$-O1(ref. 39) (headgroup flipped) or to $Q_{10}$-O4 (ref. 40), or from His59$^{NDUFS2}$ to $Q_{10}$-O1(ref. 40) only the mode with the Tyr bonded to $Q_{10}$-O4, which is consistent with mode I, has so far been observed in any structure. Importantly, this fact does not exclude other states from occurring transiently during catalysis (as low-occupancy states may be missed in structural analyses or absent under the sample preparation conditions so far tested), rather our analysis is limited to relatively stable states that accumulate to high levels. Mode I may denote both a pre-reaction state (observed without NADH, waiting for electron transfer to Q, as for the Q in CI reconstituted in proteoliposomes (PDB-8Q48)[11] and a post-reaction state (observed with NADH, waiting for protonation of the Tyr to release the $QH_2$, as for the $QH_2$ in the NADH-active-$Q_{10}$ structure). The similarity of the pre- and post-reaction states (with only the Tyr-Q hydrogen bond reversed) may lower activation energies by minimising reorganisation during the redox transition. Mode J, which lacks the Tyr hydrogen bond, may also represent both pre-reaction (sub-group J.2) or post-reaction (J.1) states: on its way into or out of Mode I the headgroup flips over and engages a non-reactive interaction with the His residue. The quinol simulations of $Q_{10}H_2$-binding presented here (Fig. 3) do not show a well-defined free energy minimum compatible with the two experimental J.1 structures (Table 2) but simulations of charge states [Tyr, His, Asp-] and [Tyr, His, AspH] show weak hydrogen bonds between the Q-methoxys, Tyr and His (Figure S5), consistent with protonation of the Tyr releasing the $QH_2$ from Mode I. Direct interactions between the Q-head and reactive groups are absent in mode K and no correlation with charge state is apparent, so mode K (only observed so far when NADH is added) is likely a post-reaction dissociative state, where quinol pauses on its way out of the channel, perhaps waiting for a conformational transition or a protonation change to release it. Figure 5 proposes a pathway for quinone reduction in the active site based on transitions between the structurally observed Q-binding modes.

**Comparison of apparent proton-transfer networks between structures**

Comparison of the proton networks in the membrane domains of different mammalian structures (Table 1) tentatively suggests the existence of different network configurations, but these differences

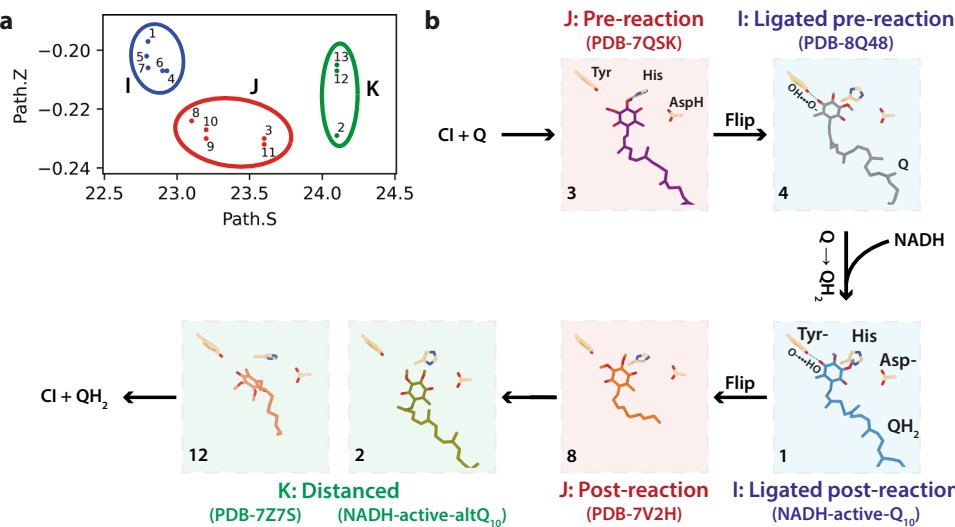

**Fig. 5 | Experimental cryoEM structures with a bound Q in three distinct binding modes. a** The relative position of the Q-head bound in the redox site is described by collective variables Path.S for its position along the channel and Path.Z for its bending and twisting. The structures in Table 2 cluster in three groups that also show distinctive hydrogen bond patterns. **b** Proposed overview reaction scheme involving closed resting and turnover states. Representative structures are shown with their model number from (**a**). The conserved active-site Tyr, His and Asp residues are shown and labelled with their protonation states, where determined from MD simulations. Q is in the oxidized state in the upper two models, and reduced ($QH_2$) in the four lower models.

should not be overinterpreted as most incomplete connections arise from unobstructed gaps, across which proton-transfer may be mediated by mobile (unresolved) water molecules. In contrast, when the gaps are physically obstructed (Table 1) then proton transfer can be blocked. As discussed above, an obstructed gap is present at ND6-TMH3 in the NADH-deactive state, which adopts a π-bulge rather than α-helix conformation. Table 1 also highlights the presence of a single obstructed gap in ND5 in every structure, either before or after TMH8-His248[ND5], consistent with its proposed switching behaviour[18,47,48]. Typically, the connection from TMH8-Lys135[ND2] to the matrix is also obstructed, with the only complete connection observed so far in *Y. lipolytica* CI[16]. In *Y. lipolytica*, mammalian Leu131[ND2] is substituted by *Yl*-Ser237, which increases hydration and supports Grotthuss-competent networks to both TMH8-Lys135[ND2] and Arg176[ND2] (*Yl*-Lys282). Mammalian Thr119[ND2] is substituted by *Yl*-Tyr225, which sits between *Yl*-Ser237 and *Yl*-Leu221 (the 'Leu gate' residue, mammalian Val115[ND2]). In *P. denitrificans*, *Pd*-Tyr231 (*Yl*-Tyr225) sits between *Pd*-Thr227 (*Yl*-Leu221) and *Pd*-Ala243 (*Yl*-Ser237), obstructing the ND2 matrix-side connection, allowing a connection from *Pd*-Thr227 to the central axis. Thus, assuming that the proton-pumping activity of the antiporter-like subunits is conserved between species, structural data suggest that the matrix half-channels in ND4 and ND5 are capable of proton uptake, but cannot yet provide any insights on whether the ND2 half-channel is active or not. It is important to recognise that the distance-based method used to evaluate the networks does not consider the detailed structure or energetics of the connections, and therefore detecting a pathway does not define it as capable of long-range proton transfer, it only highlights the possibility. Proton transfer pathways may also form transiently in states of the enzyme that have not yet been visualised. Further work is thus required to functionally evaluate candidate proton-transfer pathways identified in structural data.

## Methods

### Preparation of complex I-containing nanodiscs (CI-NDs)
Mitochondrial membranes were prepared from bovine heart mitochondria, then used to purify complex I in dodecylmaltoside (DDM) detergent[8]. The same mitochondria batch were used as used previously for cryo-EM analyses of as-prepared (oxidised) complex I[8]. The purified complex was reconstituted into phospholipid NDs containing saturating $Q_{10}$[8]. Briefly, two batches of 0.5 mg of chloroform-dissolved synthetic lipids (Avanti Polar Lipids) were prepared with total concentration 25 mg mL⁻¹ and a mass ratio of 8:1:1 of 1,2-dioleoyl-sn-glycero-3-phosphocholine (DOPC):1,2-dioleoyl-*sn*-glycero-3-phosphoethanol-amine (DOPE):18:1 cardiolipin (CDL). Each batch was mixed with 20 nmol of chloroform-dissolved $Q_{10}$ (i.e., 40 nmol $Q_{10}$ per mg lipid), the chloroform evaporated off under a stream of $N_2$, and any residual chloroform removed under vacuum for at least 2 h. The dried lipid-$Q_{10}$ mixtures were each resuspended in 457.5 µL 10 mm MOPS (pH 7.5 at 4 °C), 50 mM KCl by vigorous vortexing, and sonicated with 42.5 µL of 20% w/v sodium cholate in an ultrasonic bath (Grant Instruments Ltd., Cambridge) for 10 min. The samples were centrifuged at 7000 x *g* for 10 min, then the supernatants incubated on ice for 10 min. MSP2N2 was prepared[8] and combined with bovine complex I and the lipid-$Q_{10}$ mixture at a molar ratio of 400:10:1 lipid:MSP2N2:-complex I. Each sample was then diluted 2-fold with 0.5 mL 10 mM MOPS (pH 7.5 at 4 °C), 50 mM KCl, incubated on ice for 20 min., and run over a PD10 desalting column (Cytiva) at 4 °C. The eluates were combined, concentrated using a 100 kDa MWCO Amicon® Ultra concentrator (Merck Milipore Ltd.) to ~100 µL, and filtered using a 0.22 µm Corning® Costar® Spin-X® plastic centrifuge tube filter (Merck Milipore Ltd.). The concentrate was applied to a Superose 6 increase 5/150 column and the peak fraction containing CI-NDs eluted in 10 mM MOPS (pH 7.5 at 4 °C), 50 mM KCl. The most concentrated fractions from the monodisperse peak were used for grid preparation.

### Characterisation of CI-NDs
The CI-ND sample used for cryo-EM was characterised using the same approaches as previously[8]. Complex I and MSP2N2 concentrations were calculated by comparing data from the Pierce™ bicinchoninic acid (BCA) assay with data from the complex I NADH:APAD⁺ assay, quantified relative to a detergent-solubilised sample of known concentration[8]. Phospholipid contents were determined[8] assuming an average molecular weight of 771.6 g mol⁻¹, and that 1 mg of phospholipid occupies ~1 µL[55]. $Q_{10}$ contents were quantified by HPLC[8] and defined relative to the phospholipid phase volume. The sample used for cryo-EM contained 5.1 mg-protein mL⁻¹ (4.3 mg-CI mL⁻¹) and an average of 272 phospholipids and 3.02 $Q_{10}$ per CI-ND, similar to the sample characterised previously[8].

Catalytic activity assays were conducted at 32 °C in 96-well plates using a Molecular Devices Spectramax 384 plus plate reader ($\varepsilon_{340\text{-}380} = 4.81$ mM$^{-1}$ cm$^{-1}$), initiated by addition of 200 µM NADH. Isolated complex I and CI-ND samples were diluted to 0.5 µg mL$^{-1}$ in 20 mM Tris-HCl (pH 7.5 at 32 °C), and NADH:decylubiquinone (dQ) oxidoreductase activity assays performed with 200 µM dQ, 0.15% asolectin/CHAPS, and 10 µg mL$^{-1}$ alternative oxidase (AOX)[56]. The sample used for cryo-EM had an activity of $10.3 \pm 0.2$ µmol min$^{-1}$ mg$^{-1}$. The complex I active/deactive ratio was determined using N-ethylmaleimide (NEM): samples were either incubated on ice or deactivated by incubation at 37 °C for 15 min (ref. 28), then treated with either 2 mM NEM (200 mM stock in DMSO) or an equivalent volume of DMSO, and incubated on ice for 20 min before determining the NADH:dQ oxidoreduction activity. The proportion of active enzyme was calculated by comparing the background-subtracted specific activities of as-prepared samples ± NEM treatment with the specific activity of NEM-treated, deactivated CI-ND ($1.0 \pm 0.1$ µmol min$^{-1}$ mg$^{-1}$) was used as the background rate.

### Cryo-EM grid preparation and image acquisition

UltrAuFoil gold grids (0.6/1, Quantifoil Micro Tools GmbH)[57] were glow discharged at 20 mA for 90 s, incubated in a solution of 5 mM 11-mercaptoundecyl hexaethyleneglycol (TH 001−m11.n6-0.01, ProChimia Surfaces) in ethanol for two days in an anaerobic glovebox, then washed with ethanol and dried just before use[28]. Each grid was placed into an FEI Vitrobot™ Mark IV (Thermo Fisher Scientific) set to 100% relative humidity and 4 °C, then, immediately before application to the grid, 0.3 µL of 100 mM NADH [freshly prepared in MOPS-KOH (pH 7.5) and 50 mM KCl] was added to 3 µL of 5.1 mg mL$^{-1}$ CI-ND solution using a Hamilton® syringe (Merck) to a final concentration of 10 mM NADH, and mixed. 2.5 µL of the NADH-CI-ND solution were applied to the grids before blotting for 10 s at force setting -10 and plunge-freezing into liquid ethane. The time between addition of NADH and freezing was 32−38 s (mean 35 s). Grids were clipped using an AutoGrid clipping station (Thermo Fisher Scientific) and screened for particle number and distribution. Cryo-EM data were collected on a single grid at the UK National eBIC (Electron Bio-Imaging Centre) at the Diamond Light Source (proposal number BI22238-37) using a Gatan K3 detector and a post-column imaging energy filter (Gatan BioContinuum) operating with zero-loss filtering with a slit width of 20 eV mounted on a 300 keV Titan Krios microscope (Krios II) with fringe-free imaging (FFI), a 100 µm and 70 µm objective and C2 apertures, respectively, and EPU v. 2.12. Data were collected in super-resolution electron counting mode at a super-resolution pixel size of 0.536 Å pixel$^{-1}$ (81,000x nominal magnification) with a defocus range −0.9 to −2.3 in 0.1 µm intervals, and the autofocus routine run after recentring. AFIS was used for data acquisition. The dose rate was 18.8 electrons Å$^{-2}$ s$^{-1}$, with 2.14 s exposures captured in 40 frames. The total dose was thus 40.29 electrons Å$^{-2}$. AutoStigmate and AutoComa were used, but not AutoCTF. A total of 13,056 movies were retrieved as non-gain-corrected LZW-compressed tiff movie stacks.

### Cryo-EM data processing

The NADH-CI-ND dataset was processed using RELION 3.1.0 (ref. 58) unless stated otherwise (Supplementary Fig. 1), using a data processing pipeline as close as possible to that for the oxidised CI-ND dataset[8]. The micrographs were motion-corrected using RELION's implementation of motion correction with 5 × 5 patches, and CTF estimated using CTFFIND-4.1 (ref. 59) with an amplitude contrast of 0.1 and *ResMax* set to 4 Å. Micrographs were filtered to remove those with a *rlnCtfFigureOfMerit* value outside the range 0−6, and ice-contaminated micrographs removed manually to give a total of 12,559 micrographs (from 13,056). 2,721,931 particles were picked using RELION's *AutoPicking* tool with a 3D map input [Electron Microscope Data Bank (EMDB) ID: 14127] low-pass filtered to 20 Å. The *AutoPicking* parameters were picking

threshold = 0.2, minimum inter-particle distance = 140 Å, maximum standard deviation noise = 1.2, minimum average noise = −0.5.

Particles were extracted with an initial 4.5x downscaling to 2.412 Å pixel$^{-1}$ then randomised and split into four equal-size sets. Each set was 3D refined without a mask or solvent flattening, but with the argument *--maxsig 2000* to limit the maximum number of poses and translations considered. The four sets were then subjected to 3D classification with local angular search into five classes (angular sampling 1.875°) to remove non-complex I classes. A total of 831,889 particles were then re-extracted with re-centring at the nominal pixel size (1.072 Å pixel$^{-1}$; 2x bin), randomised and split into two equal sets. Following another round of 3D refinement and 3D classification with identical settings, to remove aberrant particles, 733,866 particles were retained (from 12,557 micrographs). AFIS compensation scripts were executed using 45 clusters (or *OpticsGroups*). The particles were subjected to iterative rounds of CTF refinement[58], to estimate anisotropic magnification, beam tilt, trefoil, 4$^{\text{th}}$ order aberration, and per-particle defocus, astigmatism and *B*-factor parameters, then to Bayesian polishing, after which CTF refinement was reiterated.

The particles were 3D refined with a mask [from PDB-7QSM][8] using RELION *MaskCreate*] and with solvent flattening to give a global resolution of 2.14 Å [at Fourier shell correlation (FSC) = 0.143], equivalent to the Nyquist resolution at the nominal 1.072 Å pixel$^{-1}$ size. Signal subtraction was then performed to remove most of the non-complex I contribution (i.e., the MSP2N2s and ND bilayer) using the mask. Following 3D classification (six classes; local angular search with sampling at 1.875, 0.9375, 0.46875, and 0.234375°), the retained 732,283 particles were re-grouped into 6,585 groups for robust sigma-noise and intensity scale-factor estimates. A second round of 3D classification (six classes; local angular search with angular sampling at 7.5, 3.75, 1.875, 0.9375, 0.46875, and 0.234375°) was then performed to separate the particles into the active, deactive and slack states. By map-to-map comparisons of their global conformations with corresponding oxidised bovine CI-ND structures[8], class 3 (136,244 particles) was assigned to complex I in the 'active' resting state, classes 1, 2, 5 and 6 (521,652 particles, combined following detailed investigation of conformation and map features) to complex I in the 'deactive' resting state, and finally class 4 (75,970 particles) to the slack state[8]. The active, deactive and slack classes were re-grouped to 2595, 2618, and 2587 groups, respectively then signal reverted to include the ND densities, and repolished at 0.75375 Å pixel$^{-1}$ (1.4x bin). The active, deactive, and slack classes refined to 2.30, 2.01, and 2.47 Å resolution, respectively.

### Focused classifications and generation of final maps

All further processing was carried out at the nominal pixel size of 0.75375 Å pixel$^{-1}$. A subtract-focus-classify-revert method employed previously for CI-ND datasets[8] was implemented to delineate any heterogeneity present in the substrate-binding sites (Supplementary Fig. 1). Briefly, individual classes were first subject to signal subtraction to retain only the hydrophilic arm of complex I using a mask generated from $Q_{10}$ modelled in the oxidised CI-ND active-$Q_{10}$ model (PDB-7QSK)[8] for NADH-active, DDM modelled in the oxidised CI-ND deactive-ligand model (PDB-7QSM)[8] for NADH-deactive, and cholate modelled in the oxidised CI-ND slack model (PDB-7QSO)[8] for NADH-slack, focus-refined, and then focus-classified (3D classification without alignment (regularisation parameter, $T = 100$)). Masks for focused classification were generated using the *molmap* function in UCSF ChimeraX[60] followed by RELION *MaskCreate* with the following parameters: low-pass filter = 4 Å (equivalent to the molmap resolution), binarization threshold = 0.21, pixel extension = 0, soft cosine edge pixel extension = 2 (or 5 for the second focused classification of NADH-active, Supplementary Fig. 1).

For the active state, a $Q_{10}$-bound sub-state 'NADH-active-$Q_{10}$' (45,337 particles) was obtained in the first round of focus-classification, and a DDM-bound sub-state 'NADH-active-DDM' (29,360 particles) and another $Q_{10}$-bound sub-state 'NADH-active-alt$Q_{10}$' (61,547 particles) in

the second round (Supplementary Fig. 1). For the deactive state, focused classification was unsuccessful, and therefore it was kept as a single class 'NADH-deactive-DDM' (521,652 particles) (Supplementary Fig. 1). For the slack state, focused classification similarly did not lead to sub-states with distinguishable traits, and therefore it was kept as a single class 'NADH-slack' (75,970 particles) (Supplementary Fig. 1). The identified sub-states were signal reverted and then 3D refined with an appropriate global mask (see below) to give the global map, and the global resolution estimated from the FSC between two independent, unfiltered half-maps (FSC = 0.143) (Supplementary Table 1 and Supplementary Fig. 2).

The model-generated masks used for 3D refinement with solvent flattening and resolution estimation were generated from the oxidised CI-ND models (active, PDB-7QSK; deactive, PDB-7QSM; slack, PDB-7QSO)[8] in UCSF ChimeraX[60] using the *molmap* function, before being low-pass filtered to 15 Å and having a 6-pixel soft cosine edge added using RELION *MaskCreate*. All consensus maps were locally sharpened from the unsharpened, unfiltered half-maps generated from RELION using *phenix.autosharpen* in Phenix 1.18.2–3874 (ref. 61), setting the resolution limit to the highest local resolution determined from RELION *LocalRes* (Supplementary Table 1 and Supplementary Fig. 2), and with a local sharpening box size of 15x15x15 pixels and a targeted overlap of 5 pixels. The Phenix *Autosharpen* real-space maps are normalised (mean ∼ 0; standard deviation/RMS ∼ 1) and contoured in standard deviation (σ) units. The locally sharpened global maps were used for model building and refinement (Supplementary Table 1). Mollweide projections were plotted using *Python* and *Matplotlib* to visualise orientational bias, and the directional resolution anisotropy calculated using *3DFSC*[62] (Supplementary Fig. 2).

To generate smoothened cryo-EM densities for MSP2N2 helices suitable for visualisation in Fig. 1a, a Gaussian filter was applied using the *Map Filter* function in UCSF ChimeraX[60], with a width (standard deviation of the 3D isotropic Gaussian function in the physical distance units of the map) of 1.5 Å.

## Model building, refinement and validation

Published bovine CI-ND models [PDB IDs: 7QSK (active) or 7QSM (deactive)][8] were rigid-body fitted into respective active/deactive/slack maps using the *Fit in Map* tool in UCSF ChimeraX[60], and *Curlew* all-atom-refined using Coot 0.9.6.2-pre[63] following the generation of ligand restraints and addition of hydrogen atoms using the *eLBOW* and *ReadySet!* tools, respectively, in Phenix 1.18.2-3874 (ref. 61). Atom resolvabilities (Q-scores) in the respective cryo-EM maps were calculated using MapQ[64], and any sidechain and/or backbone outliers identified and corrected in Coot. New resolvable regions, including post-translational modifications and extensions at chain termini, were built manually in Coot. As reported previously[8], the two domestic cow hearts used in this study have polymorphisms at residue position 255 of NDUFA10 (Lys) and 129 of NDUFS2 (Arg).

Densities for phospholipid molecules were identified with the *Unmodelled blobs* tool in Coot. All non-cardiolipin phospholipids were modelled as phosphatidylethanolamines (monomer code: 3PE) unless density features indicated phosphatidylcholine (PC1) to be more likely. Phospholipid tails were clipped where necessary using the delete tools in Coot and PyMOL 2.4.1[65]). Where unambiguously identified, substrates/ligands such as $Q_{10}$ (U10), NADH (NAI), cholate (CHD), DDM (LMT), and 2'-deoxyguanosine-5'-triphosphate (dGTP; DGT), and amino acid modifications including (S)-γ-hydroxyarginine (WYK)[10] were also modelled, and ligand restraints generated accordingly using the *eLBOW* tool in Phenix 1.20.1-4487[61]. Water molecules were placed into density peaks identified with the *Find Waters* function in Coot, with the distance to protein atoms set to 2.4-3.4 Å. The identified waters were manually edited to remove falsely placed waters (based on hydrogen bonding geometries, strength and shape of densities, and steric clashes) and bulk solvent waters, and to

add waters missed due to uncertain positions of surrounding side-chains or waters.

The manually inspected models were then real-space refined against the respective locally sharpened consensus maps in Phenix 1.20.1-4487 (ref. 61) with default Ramachandran restraints. The *rotamers.fit* option was occasionally set to *outliers* (default = *outliers_or_poormap*) to remove genuine rotameric errors. This real-space refinement step was performed iteratively with manual adjustments in Coot to correct outliers. Custom geometry restraints were imposed during Phenix real-space refinement, and no secondary structure restraints were applied.

The model statistics for the active and deactive states (Supplementary Table 1) were produced by Phenix, MolProbity, and EMRinger. Model-to-map FSC curves were generated using *phenix.validation_cryoem* in Phenix 1.20.1-4487. Hydrogen bonding contacts within individual CI-ND models (with hydrogens added using *phenix.ready_set* and/or *phenix.reduce*[61]) were identified using the *hbonds* command in UCSF ChimeraX[60].

## Molecular dynamics simulations

The simulation model is based originally on the structure of complex I from *Mus musculus* in the closed state (PDB-6ZR2)[30]. Although the current NADH-active-$Q_{10}$ structure has higher resolution, the PDB-6ZR2 model is already robust, with negligible differences in relevant side-chain positions, and it was therefore retained for consistency with our previous simulations on the corresponding oxidized structure[8]. All cofactors, post-translational modifications, and high-confidence phospholipids present in the PDB model were retained. Lipids were built with linoleoyl (L, 18:2) acyl chains. The FeS cluster N2 was simulated in the reduced state, consistent with the excess NADH concentration used in the experiment. Protonation states of sidechains were adjusted to neutral pH, except for His59$^{NDUFS2}$, His549$^{NDUFS1}$, and His42$^{NDUFB2}$, which were di-protonated (Hsp), and Glu68$^{ND3}$, Glu36$^{NDUFS5}$, Glu262$^{ND1}$, and Glu114$^{ND4}$, which were protonated. These protonation states were chosen based on PropKa[66] calculations and analysis of the chemical environment of the sidechains. The protein complex was embedded in a bilayer composed of a mixture of 368 DLPC, 294 DLPE, 96 CDL (di-anion), and 22 oxidized $Q_{10}$ molecules with 205,387 waters, 553 Na$^+$ and 348 Cl$^-$ ions, for a total of 861,976 atoms[8].

The initial simulation model, containing an oxidized $Q_{10}$ bound to the top of the redox site, was previously relaxed and equilibrated during molecular dynamics (MD) simulations totaling 740 ns[8]. Hydration of the Q-channel was verified to be stable, with 70–80 water molecules occupying the cavity and local contacts consistent with earlier structures[7–9,14,16]. Here, the bound $Q_{10}$ was changed to $Q_{10}H_2$ by manually adding two hydrogens to form the phenol groups, and the protonation states of reactive groups [Tyr, His, Asp] were adjusted to match each of the states simulated (Fig. 3). Each of these models was further subjected to 200 ns of additional MD equilibration. The simulations were stable and displayed root-mean-square deviation (RMSD) for Cα positions of chains NDUFS7, NDUFS2, and ND1 smaller than -1.5 Å relative to both the initial model (PDB-6ZR2) and the NADH-active-$Q_{10}$ current model, as well as average hydration numbers of reactive groups consistent with the current cryoEM models. The procedure is equivalent to that used previously[8].

All simulations were conducted with GROMACS (version 2020.3)[67] at constant temperature (310 K, with the Bussi thermostat) and pressure (1 atm, with the Parrinello-Rahman barostat), and a time step of 2 fs. Long-range electrostatics were treated with the Particle Mesh Ewald method. Interactions of the protein, cofactors, lipids, and ions were described using the all-atom CHARMM36m force field (charmm36-mar2019.ff)[68]. Water was represented by the standard TIP3P model[69]. FeS centres were described using the Chang parameters[70] and calibrated parameters were used for $Q_{10}$ and $Q_{10}H_2$ (refs. 71,72). Tyrosinate (Tyr$^-$) partial charges were adjusted

($q_{OH} = -0.80$ and $q_{CZ} = -0.20$) to reproduce its quantum-chemical interaction energy to a single water molecule, following the standard CHARMM36 practice[68].

A pathway collective variable (CV) was employed to describe the position of the Q-headgroup along the channel, as used previously[8,40]. This CV is a combination of distances between the heavy atoms in the Q-headgroup and the C$\alpha$ of residues in subunits NDUFS7, NDUFS2, and ND1 exposed to the Q-binding channel. Distances are evaluated with respect to four milestone configurations that represent progressive binding of $Q_{10}H_2$. These configurations were provided previously[8] and used here without any change. The CV has two components: the Path.S variable, which describes the Q-head location along the binding channel, and the complementary Path.Z, which describes its tilting and turning[8,40].

Well-tempered metadynamics[73] simulations were performed for each charge state, starting from a configuration taken at the end (200 ns) of each canonical MD trajectory used for equilibration and described above. Metadynamics were activated in the Path.S, Path.Z coordinates and in the His59$^{NDUFS2}$ $\chi$2 dihedral (C$_\beta$-C$_\gamma$ bond torsion), with Gaussians deposited every 500 time steps (1 ps), at an initial height of 0.6 kJ mol$^{-1}$, widths of 0.4 and 0.02 units for the CV and dihedral, respectively, and a bias factor of 15.0. Walls were included to restrict sampling at 21.0 < Path.S < 26.0 and −0.245 < Path.Z < −0.220 nm$^2$, with a force constant of 10,000 kJ mol$^{-1}$ nm$^{-1}$. Productive metadynamics simulations lasted 430–450 ns for each charge state. Convergence within ±1 kJ mol$^{-1}$ of free energy differences in the CV profile was reached after ~150 ns (Supplementary Fig. S7). The effects of metadynamics and restraints were removed by re-weighting the distribution of structural properties and averaged hydration. Metadynamics simulations were performed with the PLUMED plugin (version 2.8)[74].

### Reporting summary

Further information on research design is available in the Nature Portfolio Reporting Summary linked to this article.

## Data availability

The cryoEM data generated in this study have been deposited in the electron microscopy databank (EMDB) and protein databank (PDB) with the following accession codes: EMD-55030 and PDB-9SMF (NADH-active-$Q_{10}$), EMD-55031 and PDB-9SMG (NADH-active-alt$Q_{10}$), EMD-55032 and PDB-9SMH (NADH-active-DDM), EMD-55033 and PDB-9SMI (NADH-deactive-DDM), EMD-55034 (NADH-slack). The cryo-EM raw images are available from EMPIAR with the access codes EMPIAR-13115 (NADH-CI-ND), and EMPIAR-13132 (oxidised CI-ND). Initial configurations from the metadynamics simulations are available online (https://doi.org/10.5281/zenodo.17740533). All data needed to evaluate the conclusions in the paper are present in the paper and/or the Supplementary Information. Source Data are provided as Source Data file. Source data are provided with this paper.

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

## Acknowledgements

We thank D. Chirgadze, S.W. Hardwick and L. Cooper (University of Cambridge Cryo-EM facility) for assistance with cryo-EM grid screening; Y. Chaban (eBIC) and H. R. Bridges (MRC MBU) for assistance with the remote cryo-EM data collection during the COVID-19 pandemic with restricted access; Diamond Light Source for microscope access and support of the cryo-EM facilities at the UK National eBIC (Electron Bio-Imaging Centre) at Diamond Light Source (Harwell Science and Innovation Campus, Didcot, UK), under the proposal number BI22238-37 on Krios II, funded by the Wellcome Trust, MRC, and BBSRC; the SDumont cluster in the National Laboratory for Scientific Computing (LNCC/MCTI, Brazil) for computational resources; and A. Raine and A. J. Nelson (MRC MBU) for IT support. This work was supported by the Medical Research Council (MC_UU_00015/2 and MC_UU_00028/1 to J.H.) and Fundação de Amparo à Pesquisa do Estado de São Paulo (FAPESP, grant 2023/00934-5 to G.M.A. and fellowship 2020/14542-3 to C.S.P.).

## Author contributions

I.C., J.H., and G.M.A. conceived the project. I.C. performed CI-ND reconstitution, collected and processed cryo-EM data, carried out structure model building, analyses, and interpretations, and prepared figures. I.C. and J.J.W. carried out biochemical and kinetic characterisations. J.J.W. prepared complex I for reconstitution, assisted by I.C.; C.S.P. and G.M.A. performed and analysed molecular simulations. J.H. supervised the project and contributed to interpretation of the data. I.C., G.M.A., and J.H. wrote the paper with input from all authors.

## Competing interests

The authors declare no competing interests.
