## [Transparent Peer Review file · Nature Communications]

Post-catalysis structures of mitochondrial complex I with ubiquinol-10 bound in the active site

Corresponding Author: Professor Judy Hirst

Version 0:

Reviewer comments:

Reviewer #1

(Remarks to the Author)

In their manuscript "Post-catalysis structures of mitochondrial complex I with ubiquinol-10 bound in the active site" Chung et al. report the structure of mammalian mitochondrial complex I reconstituted in lipid nano-discs after reduction of the complex with NADH. Within the active state of the complex they observe two binding modes of Q10 and argue that these correspond to the reduced Q10H2 molecule in different poses that correspond to transit of Q10H2 within the tunnel. They use molecular dynamics simulations to propose specific charge states of key active site residues and compare structures to identify proton-transfer control points within the hydrophilic axis of the membrane arm. Overall the paper is well written and presents new data and analysis that should be of interest to the complex I field as well as others working on the structure and mechanism of redox-coupled proton pumps.

Strengths

The main strengths of the manuscript are the clear organization and framing of the results and comparisons to known structures. Specifically, Table 1 and Supplementary figure 6 are laid out in a manner that will be beneficial to the field and provide a framework for future analysis of complex I structures. Supplementary figure 6 (with minor adjustments see below) is central to the story of the manuscript and the authors should consider swapping it with main text Fig. 4, to place in the main text of the manuscript instead of the supplement.

Weaknesses

An overall weakness of the manuscript is the discussion of how ordered water molecules with the Q tunnel may contribute to the mechanism. For example, the authors state "The Q10H2-O4 hydroxy is assumed to have accepted its proton from Tyr108NDUFS2, to which it remains strongly hydrogen bonded in the tyrosinate anion (Tyr⁻) form, whereas the identity of the direct proton donor to Q10H2-O1, usually assumed to be His59NDUFS2 is not defined by our structure." But the structure does show a network of ordered water molecules that hydrogen bond between Q10-O1 and Asp160 NDUFS2. How were these water molecules considered for the MD simulations and do the simulations recapitulate these ordered waters? Besides mentioning that the TIP3P model was used for the waters, details are missing from the methods regarding how the model was hydrated prior to the simulations.

The authors propose from the results of the MD simulations that the observed Q10H2 is hydrogen-bonded to the Tyr108NDUFS2 tyrosinate in the [Tyr⁻, His, Asp⁻] state. Given the pH 7.5 of the sample, it is reasonable to ask why the tyrosinate has not been re-protonated? This suggests a very large change in pKa (~4 pH units) of this tyrosine in the reduced state of the enzyme. What possible, if any, Grotthuss competent pathways can be seen connecting the Q-site to the bulk, if the Q-site is fully closed off from the bulk, why wouldn't any of the many water molecules observed in the Q-cavity, including some that appear to be directly hydrogen bonding to the tyrosine, donate a proton to it?

Minor

- Given that the "the global resolutions up to 2.0 Å" in the abstract refers to the Deactive state of the enzyme which does not represent the key findings in the manuscript, which are from the Active state structures that refine to global resolutions of 2.4-2.6 Å, the authors should rewrite or remove that statement as it could be misleading.
- Page 5, "NADH in the NADH-binding site", sentence 3, the authors state "Phe73NDUFV1 has undergone a dramatic rotameric shift." What makes a rotameric shift "dramatic"? If it is the distance moved, it would be best just to report the distance and avoid value-laden characterizations.

- Fig. 2 The histidine numbering is inconsistent mainly being referred to as His59NDUFS2 but in panels c and d being shown as His58NDUFS2.
- Fig. 2, Map thresholds should be defined. Is it rmsd from a normalized map? Until what contour threshold is the Q10 density clearly defined?
- On Page 6 the authors state “If the Q10H2 headgroup in NADH-active-Q10 is flipped over to match the oxidised active-Q10 structure⁸, the two methoxy groups no longer fit well into the density, the interaction between Q10H2-O4 and Tyr108NDUFS2-O is lost, and the Q10H2-O1 involved in a hydrogen bonding network with surrounding waters is replaced by the 5-methyl; His59NDUFS2 remains too distant from Q10H2-O1 (~5.0 Å) for a viable hydrogen bond.” This would be good to show in a supplemental figure.
- Page 8, the authors state “Consistently, the H220F mutation in *P. denitrificans* abolished CI catalysis⁴⁴, whereas the K237A and K237Q mutations in *E. coli* and *P. denitrificans* had only partial effects.” I assume they are referring to the residues in the ND4 subunit but this should be made explicit.
- Supplementary Fig. 6, please check that the residue labels (names and lines) are well aligned. See 6a H248ND5-TMH8 for an example of a poorly aligned set that made it difficult to track down the residue. Also, some of the residues called out in the text do not appear to be labeled in the figure, for example TMH10-Lys299ND5, this makes it difficult to use the figure to follow along with the text.
- On page 11, the authors state, “...hydrogen bonds to the Q-carbonyls from both the Tyr and the His...” it is more or less clear that they are referring to Tyr108 NDUFS2 and His59NDUFS2 or their equivalents in other structures, but it is important to use consistent nomenclature throughout to avoid possible confusion.

Reviewer #2

(Remarks to the Author)

Overview

The manuscript aims to fill important knowledge gap regarding the mechanisms of Complex I for ubiquinone-10 binding poses and proton transfer using single-particle cryogenic electron microscopy (cryo-EM) in combination with molecular dynamics (MD) simulations. The experimental design, which uses a nanodisc-reconstituted Complex I with excess NADH, enables the capture of post-catalysis structural snapshots in vitro. The manuscript is well-written with a logical flow, detailed experimental parameters, and complete comparative analyses. The findings are expected to provide valuable mechanistic insights into Complex I structure and function.

Minor Concerns

1. (Results | Page 5 | Line 4) The word “resolution” should be pluralized. Suggest: “..., except for in their resolutions ...”
2. (Results | Page 5 | Line 11) The observation of the reported “~0.9 Å” shift seems not valid for the map that was resolved at ~2 Å resolution. Suggest: “... and residues 68-72 in NDUF1 are slightly displaced. ...”
3. (Results | Page 5 | Line 29) Please provide model-density fittings for DDM in the [NADH active-DDM] and [NADH-deactive-DDM] structures in the Supplementary Figures.
4. (Results | Page 5 | Line 32) It is recommended to show a map-model correlation metric (cross-correlation coefficient) between Q10 and DDM in the binding site, respectively, to demonstrate that the densities present a mixture of the two.
5. (Results | Page 5 | Line 34) The statement that “DDM has a higher affinity for the reduced enzyme” requires further support. Please cite comparable observations from other structural studies if any. It is also not clear why DDM would respond to the reduction of the terminal FeS cluster (N2) since the binding of DDM interrupts electron transfer (but may work in stabilization of the locally reduced environment).
6. (Results | Page 7 | Line 8) A brief discussion on why a DDM was observed binding in the closed state would be helpful, in particular since this is the first structure reported in the study.
7. (Discussion | Page 11 | Line 11) Please cite the statement “... the reduction state of the system may be altered by the cryo-EM electron beam, blurring the differences between ‘oxidized’ and ‘reduced’, ...”.
8. (Methods | Characterization of CI-NDs | Paragraph 2 | Page 21) It would be helpful to comment on the correlation of the CI activity measured with the resulting cryo-EM structural analysis. Do the observed particle populations correspond to the measured functions?
9. (Methods | Cryo-EM grid preparation and image acquisition | Line 15) The “zero-energy mode” should re-phrase to “zero-loss filtering”. Suggest: “... operating with zero-loss filtering with a slit width of 20 eV ...”
10. (Methods | Cryo-EM data processing | Paragraph 3 | Line 7) The statement “... the retained 732,283 particles were re-grouped into 6,585 groups. ...” should be clarified – Was this clustering based on k number given or the image intensity scaling?
11. (Methods | Focused classifications and generation of final maps | Paragraph 1 | Line 11) Please clarify the statement of “equivalent to the molmap resolution”. Also, provide a unit for the low-pass filter used in data processing.
12. (Deposition ID D_1292150746 and PDB 9MSF, 9SMH, and 9SMG) In the PDB validation reports, the unmasked FSC (orange) curves drop below 0.50 at mid-resolution range (about 8 Å⁻¹) and then rise again (about 4.5 Å⁻¹). This may implicate an improper global mask applied or biased particle orientations during image reconstruction. If the deposited maps were generated by focused refinement, please document this and provide the masks used in the Methods or Supplementary sections.

Reviewer #3

(Remarks to the Author)

The manuscript by Chung et al. combines high resolution cryo electron microscopy and computer simulations of respiratory

complex I, a large proton pump in the inner mitochondrial membrane. They have compared the current results in which complex I is reduced by NADH to their previous study and suggest that the bound quinone molecule is Q10H2. Even though the structural data are interesting, it lacks novel insights. Authors claim they have trapped Q10H2, but the evidence in favour of this is not strong. Authors say there are no major differences with and without NADH, whereas the previous structures of porcine complex I (<https://doi.org/10.1038/s41594-022-00722-w>) have shown differences in NADH bound and unbound states. It appears even though NADH may be bound, the reduction of Q10 may not have substantially occurred in these experimental conditions and the state they have observed when Q10 molecule is hydrogen bonding to tyrosine may largely consist of an oxidized Q10. On the other hand, the alternative Q10 positions seems more likely to be a Q10H2 state. Structure and simulation studies (DOI: 10.1126/science.125985, <https://doi.org/10.1073/pnas.1805468115>, <https://doi.org/10.1021/jacs.9b13450>, <https://doi.org/10.1073/pnas.1503761112>) have clearly supported a weaker binding of Q10H2 and its stable position further away from the binding site closest to N2 FeS cluster. Also, a longer binding of Q10H2 in hydrogen bonding to tyrosine and close to FeS cluster N2, runs a risk of running reaction in reverse.

It is unclear to this referee that why the computer simulations are performed on an older structure (PDB ID 6ZR2) when the authors have an improved resolution structure such as in this manuscript available. This raises serious concerns about the conclusions drawn from the joint structural and simulation analysis. Authors could have tested on the current structure if it is Q10H2 or Q10 in the two conformations resolved in the structure. This is the strength of computer simulations, especially when combined with structural biology experiments. The computer simulation results have strong dependence on starting coordinates, conditions and quality of the structure, but here structural insights from a higher resolution structure (2 Å) are forcefully mixed with simulations on an older 3.1 Å structure.

Minor points - authors did not model semiquinone radical species or quinol anion, which have been considered functional species in complex I. Also, how were the parameters of deprotonated tyrosine constructed?

A hydrogen bonded connection from the Q10 head group to the membrane domain can be seen in the structure. Does this mean protons can travel along this entire path diffusing across subunits.

Version 1:

Reviewer comments:

Reviewer #1

(Remarks to the Author)

The authors have satisfactorily addressed all of my concerns. The manuscripts clarity and discussion around limitations have been improved and I support publication.

Reviewer #2

(Remarks to the Author)

I have reviewed the revised manuscript and find that the authors have adequately addressed all my comments.

Reviewer #3

(Remarks to the Author)

Thank you. Authors have satisfactorily answered several of my comments. However, the main issue of simulating a structure to explain the experimental data on another structure that too with a vastly different resolution remains. As authors themselves repeatedly said, their results derive by comparing the simulation data with the experimental data, but this is not done. The simulation data is on a lower resolution coordinates, whereas the experimental data is based on a much higher resolution structure. I do not intend to hinder the publication of their work, but given that they have utilized the metadynamics approach, which is a reasonably fast method, performing similar

simulations (at the least) in minimal states on the new structure using similar model construction approach should not be hard to do. The successful results will then confirm their claim and conclusions on the presence of ubiquinol in their new structural preparation, which experimentally suggests the presence of ubiquinol formation. Furthermore, it is not about simulation sampling convergence, but to do with the starting coordinates. As authors response points to, it is a given that a lower resolution cryo-EM structure has less accurate amino acid sidechain packing than the higher resolution one. This difference can easily lead to different simulated outcomes, unless shown otherwise. This is why consolidating their excellent structural data with simulations on higher resolution data even for selected protonation states will also go far and beyond.

Minor points

I agree it is known from experiments that semiquinone may not populate in complex I catalysis (even though it must form) and therefore this possibility can be discarded. However, there are several reports on deprotonated ubiquinol (anion) in complex I catalysis. See for instance, <https://doi.org/10.1002/1873-3468.14518>; <https://doi.org/10.1101/2024.09.06.611712>; <https://doi.org/10.3389/fchem.2021.672969>.

It would be good to consider this possibility (even if not supported by computer simulations) because it is possible that their NADH-active-Q10 and NADH-active-altQ10 states may consist of protonated tyrosine strongly hydrogen bonding with the negatively charged ubiquinol anion, which has formed after proton abstraction from histidine 59 and aspartic acid 160.

Authors in their revised version of the manuscript wrote "The average hydration of reactive groups in this charge state (Tyr coordinated to ~3 waters, His to ~3 waters, and Asp to 4–5 waters) closely matches the hydration observed in the NADH-active-Q10 model (Figure 2a)."

Can authors elaborate how is this average hydration shown (in Figure 2a)? Alternatively, this could be shown separately in supplementary information for example by overlaying the simulated density of water with the structural water molecules.

Thank you for the constructive and helpful reviews of our manuscript entitled “Post-catalysis structures of mitochondrial complex I with ubiquinol-10 bound in the active site”. We have now addressed the comments from the reviewers as detailed below and amended our manuscript accordingly (as requested, the relevant sections of text are highlighted in colour).

We trust that you will now find our manuscript suitable for publication and look forward to hearing from you further.

Best wishes,
Judy Hirst (on behalf of all authors)

Reviewer #1

In their manuscript “Post-catalysis structures of mitochondrial complex I with ubiquinol-10 bound in the active site” Chung et al. report the structure of mammalian mitochondrial complex I reconstituted in lipid nano-discs after reduction of the complex with NADH. Within the active state of the complex they observe two binding modes of Q10 and argue that these correspond to the reduced Q10H2 molecule in different poses that correspond to transit of Q10H2 within the tunnel. They use molecular dynamics simulations to propose specific charge states of key active site residues and compare structures to identify proton-transfer control points within the hydrophilic axis of the membrane arm. Overall the paper is well written and presents new data and analysis that should be of interest to the complex I field as well as others working on the structure and mechanism of redox-coupled proton pumps.

We thank the reviewer for their positive evaluation of our manuscript.

Strengths

The main strengths of the manuscript are the clear organization and framing of the results and comparisons to known structures. Specifically, Table 1 and Supplementary figure 6 are laid out in a manner that will be beneficial to the field and provide a framework for future analysis of complex I structures. Supplementary figure 6 (with minor adjustments see below) is central to the story of the manuscript and the authors should consider swapping it with main text Fig. 4, to place in the main text of the manuscript instead of the supplement.

We thank the reviewer for their enthusiasm about our analysis, as presented in Table 1 and Suppl. Fig. 6. While we agree the information in Suppl. Fig. 6 is central to our story, we chose to place it in the supplementary section because we have already shown matching analyses of earlier (albeit lower resolution) structures in earlier papers (notably, Grba et al. 2023 [ref. 10]). In addition, we feel that the very detailed and granular information shown in the figure for specific states is better packaged for evaluation and interpretation in Table 1, as it allows us to take our analysis to the next level, comparing the results of the same analysis repeated on different states and structures. The information in Figure 4 also represents a further development of previous analyses, which did not consider the access of protons through the supernumerary structures. We therefore thank the reviewer for their suggestion, but on balance, we prefer to retain the current arrangement, to ensure focus on the newest and most innovative aspects of our current work.

Weaknesses

An overall weakness of the manuscript is the discussion of how ordered water molecules with the Q tunnel may contribute to the mechanism. For example, the authors state “The Q10H2-O4

hydroxy is assumed to have accepted its proton from Tyr108NDUFS2, to which it remains strongly hydrogen bonded in the tyrosinate anion (Tyr⁻) form, whereas the identity of the direct proton donor to Q10H2-O1, usually assumed to be His59NDUFS2 is not defined by our structure.” But the structure does show a network of ordered water molecules that hydrogen bond between Q10-O1 and Asp160 NDUFS2.

We agree – the fact that the Tyr remains bonded to the nascent quinol makes their interaction (as a proton donor/acceptor pair) clear, whereas our data do not confirm the origin of the proton on quinol O1. Although we (and the field) expect that the essential His59 residue is most likely the second proton donor (ie. the quinol has dissociated from it in our structures, following the transfer), consistent with a role for the His in the energy conversion mechanism, an alternative is that proton transfer is simply water mediated. We now say, to explicitly note this (page 8) “[The donor] is usually assumed to be His59^{NDUFS2}, in which case a productive interaction between the Q₁₀H₂-O1 and one of the two His-N centres must have formed and broken before the formation of the states observed here. Alternatively, we cannot exclude that the proton is transferred by water molecules, via the hydrated network that is observed between O1, His59 and Asp160.”

How were these water molecules considered for the MD simulations and do the simulations recapitulate these ordered waters? Besides mentioning that the TIP3P model was used for the waters, details are missing from the methods regarding how the model was hydrated prior to the simulations.

Yes, the MD simulations recapitulate the hydration observed in the structures. We have now added a sentence to the Results (page 7) to note the average number of water molecules bound to the side-chains of the His/Tyr/Asp residues adjacent to the bound quinol, for comparison with the cryo-EM models. Further information has also been added to the methods on hydration of the model as requested (page 26). Note that no special hydration procedure was required during model construction. Water penetration into and exchange from the amphipathic Q-channel occurs on the tens-of-nanoseconds timescale (see for example Fig. S8 in ref. 40). Consequently, the long equilibration phase used for model building (740 ns of MD simulation) is sufficient to establish a stable and experimentally consistent hydration pattern in the Q-channel of the simulation models.

The authors propose from the results of the MD simulations that the observed Q10H2 is hydrogen-bonded to the Tyr108NDUFS2 tyrosinate in the [Tyr⁻, His, Asp⁻] state. Given the pH 7.5 of the sample, it is reasonable to ask why the tyrosinate has not been re-protonated? This suggests a very large change in pKa (~4 pH units) of this tyrosine in the reduced state of the enzyme. What possible, if any, Grothuss competent pathways can be seen connecting the Q-site to the bulk, if the Q-site is fully closed off from the bulk, why wouldn't any of the many water molecules observed in the Q-cavity, including some that appear to be directly hydrogen bonding to the tyrosine, donate a proton to it?

We propose that the tyrosinate is stabilised by direct hydrogen bonding to the quinol: the two of them share a single proton between them, and the pKa of the tyrosine should not be considered as though it exists in isolation. As there are no Grothuss competent pathways detected connecting the Q-site to the bulk, the Tyr is effectively ‘buried’ and so its re-protonation from the bulk is anticipated to be an activated process (with a free-energy barrier) – this is consistent with our assignment of a ‘pre-reprotonation’ state, in which the effect of the electron transfer has only been to reorganise the proton distribution within the cavity. We note that the protonation states of the local ‘waters’ are undefined, and that the pKa of an isolated water molecule is actually much higher than that of tyrosine. For all these reasons, and based on the results of our simulations, we believe our proposal is reasonable. In the bigger picture, how the protons are transferred, ultimately from the matrix to the nascent quinol upon electron transfer, is an area of intense

debate in complex I: under sustained turnover the waters/residues in the Q-site cannot keep on providing protons without being replenished, but whether this occurs simply by an ‘innocent’ uptake pathway or is itself a trigger for the energy-transduction mechanism is not yet known.

Minor

- Given that the “the global resolutions up to 2.0 Å” in the abstract refers to the Deactive state of the enzyme which does not represent the key findings in the manuscript, which are from the Active state structures that refine to global resolutions of 2.4-2.6 Å, the authors should rewrite or remove that statement as it could be misleading.

We note that this statement is factually correct, but have amended it as requested to note that the structures presented in our paper (including the deactive state that we analysed in detail for its connectivity) were at global resolutions of 2.0 to 2.6 Å.

- Page 5, “NADH in the NADH-binding site”, sentence 3, the authors state “Phe73NDUFV1 has undergone a dramatic rotameric shift.” What makes a rotameric shift “dramatic”? If it is the distance moved, it would be best just to report the distance and avoid value-laden characterizations.

We now say Phe73 has changed rotamer.

- Fig. 2 The histidine numbering is inconsistent mainly being referred to as His59NDUFS2 but in panels c and d being shown as His58NDUFS2.

Thank you very much for pointing out this mistake – His59 is correct and the labels on Fig 2 have now been corrected.

- Fig. 2, Map thresholds should be defined. Is it rmsd from a normalized map? Until what contour threshold is the Q10 density clearly defined?

We have added a sentence on page 24 as follows: “The Phenix *Autosharpen* real-space maps are normalised (mean ~ 0 ; standard deviation/RMS ~ 1) and contoured in standard deviation (σ) units” and the σ units are now reported in the relevant figure legends (Fig. 2, Supp. Fig. 2, Supp. Fig. 3, Supp. Fig. 4). The $Q_{10}(H_2)$ density is clearly defined and continuous (headgroup and isoprenoid tail) up to ~ 7.0 (NADH-active- Q_{10}) or ~ 3.5 (NADH-active-alt Q_{10}) σ .

- On Page 6 the authors state “If the Q10H2 headgroup in NADH-active-Q10 is flipped over to match the oxidised active-Q10 structure8, the two methoxy groups no longer fit well into the density, the interaction between Q10H2-O4 and Tyr108NDUFS2-O is lost, and the Q10H2-O1 involved in a hydrogen bonding network with surrounding waters is replaced by the 5-methyl; His59NDUFS2 remains too distant from Q10H2-O1 (~ 5.0 Å) for a viable hydrogen bond.” This would be good to show in a supplemental figure.

We now show the flipped (i.e. incorrect) $Q_{10}H_2$ conformation for the NADH-active- Q_{10} state in Supplementary Figure 3, alongside the correct conformation for comparison, and with the main challenges to this interpretation noted. The density fit for the flipped pose is less convincing than for the modelled pose, and importantly the interactions it makes are much less favourable than for the modelled pose. We describe this on page 6: “If the $Q_{10}H_2$ headgroup in NADH-active- Q_{10} is flipped over to match the oxidised active- Q_{10} structure, the two methoxy groups no longer fit well into the density, the interaction between $Q_{10}H_2-O_4$ and Tyr108^{NDUFS2}-O is lost, and the $Q_{10}H_2-O_1$ involved in a hydrogen bonding network with surrounding waters is replaced by the 5-methyl; His59^{NDUFS2} remains too distant from $Q_{10}H_2-O_1$ (~ 5.0 Å) for a viable hydrogen bond (Supplementary Figure 3).”

- Page 8, the authors state “Consistently, the H220F mutation in *P. denitrificans* abolished CI catalysis⁴⁴, whereas the K237A and K237Q mutations in *E. coli* and *P. denitrificans* had only partial effects.” I assume they are referring to the residues in the ND4 subunit but this should be made explicit.

Thank you for noting these omissions, we have now added the missing superscripts.

- Supplementary Fig. 6, please check that the residue labels (names and lines) are well aligned. See 6a H248ND5-TMH8 for an example of a poorly aligned set that made it difficult to track down the residue. Also, some of the residues called out in the text do not appear to be labeled in the figure, for example TMH10-Lys299ND5, this makes it difficult to use the figure to follow along with the text.

We have carefully reviewed all three panels in Suppl. Fig. 6 to optimise their clarity as much as possible, although noting that the extensive number of residues to label makes them challenging. We have also checked the relevant sections of our manuscript against the figures to ensure that all the residues specifically referred to are now labelled (including Lys299ND5).

- On page 11, the authors state, “...hydrogen bonds to the Q-carbonyls from both the Tyr and the His...” it is more or less clear that they are referring to Tyr108 NDUFS2 and His59NDUFS2 or their equivalents in other structures, but it is important to use consistent nomenclature throughout to avoid possible confusion.

Thank you for noting this, we have now used the complete nomenclature.

Reviewer #2

Overview

The manuscript aims to fill important knowledge gap regarding the mechanisms of Complex I for ubiquinone-10 binding poses and proton transfer using single-particle cryogenic electron microscopy (cryo-EM) in combination with molecular dynamics (MD) simulations. The experimental design, which uses a nanodisc-reconstituted Complex I with excess NADH, enables the capture of post-catalysis structural snapshots in vitro. The manuscript is well-written with a logical flow, detailed experimental parameters, and complete comparative analyses. The findings are expected to provide valuable mechanistic insights into Complex I structure and function.

We thank the reviewer for their very favourable evaluation of our manuscript.

Minor Concerns

1. (Results | Page 5 | Line 4) The word “resolution” should be pluralized. Suggest: “..., except for in their resolutions ...”

Done.

2. (Results | Page 5 | Line 11) The observation of the reported “~0.9 Å” shift seems not valid for the map that was resolved at ~2 Å resolution. Suggest: “... and residues 68-72 in NDUF1 are slightly displaced. ...”

Done.

3. (Results | Page 5 | Line 29) Please provide model-density fittings for DDM in the [NADH active-DDM] and [NADH-deactive-DDM] structures in the Supplementary Figures.

We now show additional model-density fittings for the DDMs in Supplementary Figure 3 as requested.

4. (Results | Page 5 | Line 32) It is recommended to show a map-model correlation metric (cross-correlation coefficient) between Q10 and DDM in the binding site, respectively, to demonstrate that the densities present a mixture of the two.

We apologise that this statement was unclear. A density we described in an earlier publication on the oxidised state was modelled as a mixture of Q10 and DDM – but here we are confident that the density under discussion, which is very nicely resolved, can be interpreted as DDM only. We have rephrased our text so that this is now clear (page 5).

5. (Results | Page 5 | Line 34) The statement that “DDM has a higher affinity for the reduced enzyme” requires further support. Please cite comparable observations from other structural studies if any. It is also not clear why DDM would respond to the reduction of the terminal FeS cluster (N2) since the binding of DDM interrupts electron transfer (but may work in stabilization of the locally reduced environment).

This was just a suggestion based on the higher occupancy, and we agree that it is hard to see why N2 reduction would have this effect. We have shortened our comment to now read “The higher occupancy observed here suggests that DDM might have a higher affinity for the reduced than the oxidised enzyme” accordingly.

6. (Results | Page 7 | Line 8) A brief discussion on why a DDM was observed binding in the closed state would be helpful, in particular since this is the first structure reported in the study.

The protein was purified in DDM, which must have entered the binding site and been retained, despite it being removed from solution. Most likely, as we have commented, it has also been there in earlier structures, but has been missed due to their more limited resolution.

7. (Discussion | Page 11 | Line 11) Please cite the statement “... the reduction state of the system may be altered by the cryo-EM electron beam, blurring the differences between ‘oxidized’ and ‘reduced’, ...”.

We have added a reference (ref. 53) to Hussein, R. et al. Cryo-electron microscopy reveals hydrogen positions and water networks in photosystem II. *Science* 384, 1349–1355 (2024). This paper describes high resolution (1.7 Å) data and describes changes in cluster bond lengths upon exposure, noting that “This aligns with previous observations that high valent metal centers are rapidly reduced by electron beam exposure during cryo-EM data collection...”

8. (Methods | Characterization of CI-NDs | Paragraph 2 | Page 21) It would be helpful to comment on the correlation of the CI activity measured with the resulting cryo-EM structural analysis. Do the observed particle populations correspond to the measured functions?

We noted on page 6 that the proportion of closed particles in the analysis was 18.6%, consistent with the biochemical analysis shown in Suppl. Fig. 1d. However, we realise now that we presented just the data in Suppl. Fig. 1d, without providing the derived values for comparison – and so this information and the comparison has now been added to the figure legend: “22.4 ± 2.4% of the as-prepared enzyme was found to be in the active/closed state, using the specific activity of NEM-treated, deactivated CI-ND ($1.0 \pm 0.1 \mu\text{mol min}^{-1} \text{mg}^{-1}$) as the background rate. The equivalent value calculated from the particle numbers is 20.7%.” The value measured should not have been included in the methods section and has been removed.

9. (Methods | Cryo-EM grid preparation and image acquisition | Line 15) The “zero-energy mode” should re-phrase to “zero-loss filtering”. Suggest: “... operating with zero-loss filtering with a slit width of 20 eV ...”

Done.

10. (Methods | Cryo-EM data processing | Paragraph 3 | Line 7) The statement "... the retained 732,283 particles were re-grouped into 6,585 groups. ..." should be clarified – Was this clustering based on k number given or the image intensity scaling?

We have clarified this phrase as follows: "... the retained 732,283 particles were re-grouped into 6,585 groups for robust sigma-noise and intensity scale-factor estimates."

11. (Methods | Focused classifications and generation of final maps | Paragraph 1 | Line 11) Please clarify the statement of "equivalent to the molmap resolution". Also, provide a unit for the low-pass filter used in data processing.

We now state "Masks for focused classification were generated using the *molmap* function in UCSF ChimeraX⁵⁴ followed by RELION *MaskCreate* with the following parameters: low-pass filter = 4 Å, binarization threshold..."

12. (Deposition ID D_1292150746 and PDB 9MSF, 9SMH, and 9SMG) In the PDB validation reports, the unmasked FSC (orange) curves drop below 0.50 at mid-resolution range (about 8 Å⁻¹) and then rise again (about 4.5 Å⁻¹). This may implicate an improper global mask applied or biased particle orientations during image reconstruction. If the deposited maps were generated by focused refinement, please document this and provide the masks used in the Methods or Supplementary sections.

We thank the reviewer for their comment. None of the deposited maps were generated by focused refinement. We provide six maps per deposition: (1) a locally-sharpened global map generated using Phenix Autosharpen [this is the main map]; (2) a globally-sharpened full map generated using RELION PostProcess; (3) an unfiltered, unsharpened full map generated using RELION 3D Refinement; (4-5) two unfiltered, unsharpened half-maps generated using RELION 3D Refinement; (6) a global mask used for the final RELION 3D Refinement. We do note a degree of preferred orientation or uneven angular sampling in our sample (see Supplementary Figure 2), which may be responsible for the observed phenomenon, but the directional FSCs and directional resolution anisotropy calculated using 3DFSC (see Supplementary Figure 2) suggest otherwise. It is also possible that the dip in unmasked FSC arises from heterogeneous, low-to-medium resolution signal contributions from the MSP2N2 helices, which lie outside the global complex I mask.

Reviewer #3

The manuscript by Chung et al. combines high resolution cryo electron microscopy and computer simulations of respiratory complex I, a large proton pump in the inner mitochondrial membrane. They have compared the current results in which complex I is reduced by NADH to their previous study and suggest that the bound quinone molecule is Q10H2.

Even though the structural data are interesting, it lacks novel insights. Authors claim they have trapped Q10H2, but the evidence in favour of this is not strong.

As with any ligand or substrate bound in a cryo-EM density map the identity of the species we have modelled is not 'proven'. We have assumed the bound species is Q₁₀H₂ rather than Q₁₀ because Q₁₀ has a much higher reduction potential than NADH (which is present in great excess) and is rapidly reduced by complex I. We also do not favour a semiquinone species, because these are not observed in material quantities spectroscopically. We therefore believe the assumption that the species is Q₁₀H₂ is a robust assumption.

Authors say there are no major differences with and without NADH, whereas the previous structures of porcine complex I (<https://doi.org/10.1038/s41594-022-00722-w>) have shown differences in NADH bound and unbound states.

We assume that the reviewer is referring to Figures 2 and 3 in the paper by Gu and coworkers, in which structures in the presence and absence of NADH and short-chain Q analogues (as well as the native Q₁₀) were compared. Here, it is most relevant to compare the Q₁₀-bound structures in the absence (Q₁₀ bound at 'site 1') and presence (Q₁₀ bound at 'site 2') of NADH. First, we note substantial unmodelled density in the latter case (Fig. 2k) that overlaps the Q₁₀ modelled in site 1 in the former case – this density clouds the distinction between the two states. Fig. 3a-b then shows a small shift in His92 (= His59) between these two states that the authors propose is due to the His and Tyr being deprotonated following Q reduction by NADH. However, no further evidence is provided to support this assertion (for example, molecular dynamics simulations). The authors go on to correlate the His position to its proposed protonation state (although the shift is not consistent as it is not observed in the case of Q₁ + NADH). Considering only Q₁₀, it is also possible to correlate the shift to the occupancy of site 1, with the Q₁₀ moving in to displace water molecules from there. We note that we earlier observed the His sidechain to respond to Q₁₀ occupancy in our two oxidised structures (Q₁₀ present and absent). Therefore, we regard the origin of the small shift in His92 reported by Gu and coworkers as unconfirmed. We note on page 6 that “No conformational changes are observed in the NDUFS2-β1–β2 loop that carries His59^{NDUFS2}, which was observed to move either towards (PDB-6ZKC)⁷ or away (PDB-7V2E)⁹ from Asp160^{NDUFS2} upon addition of substrates to the closed/active states of ovine and porcine complex I.”

It appears even though NADH may be bound, the reduction of Q₁₀ may not have substantially occurred in these experimental conditions and the state they have observed when Q₁₀ molecule is hydrogen bonding to tyrosine may largely consist of an oxidized Q₁₀. On the other hand, the alternative Q₁₀ positions seems more likely to be a Q₁₀H₂ state.

We added NADH for 30 s before the sample was frozen, giving more than enough time for reduction. In freeze quench spectroscopic experiments, the FMN and FeS clusters of *E. coli* complex I were fully reduced in 10 ms (De Vries et al., 2015), and it is unclear why the low potential, reduced FeS chain would not donate electrons to the nearby high potential oxidised Q₁₀. Our interpretation is further supported by molecular dynamics simulations, by their strong match to the experimental data, and we have included a thoughtful discussion in our manuscript on interpretation of Q₁₀ binding poses and the caveats associated with extant data. As noted above, we do not consider the lack of a small movement of His59 sufficient to demonstrate that the Q₁₀ has not been reduced.

Structure and simulation studies (DOI: 10.1126/science.125985, <https://doi.org/10.1073/pnas.1805468115>, <https://doi.org/10.1021/jacs.9b13450>, <https://doi.org/10.1073/pnas.1503761112>) have clearly supported a weaker binding of Q₁₀H₂ and its stable position further away from the binding site closest to N2 FeS cluster. Also, a long binding of Q₁₀H₂ in hydrogen bonding to tyrosine and close to FeS cluster N2, runs a risk of running reaction in reverse.

Thank you for bringing these references to our attention, although we assume that 10.1126/science.125985 “Planetary boundaries: Guiding human development on a changing planet” is not the reference intended. The following three references are simulations based on the structure of complex I from *Thermus thermophilus*. The first (Warnau et al.) considers binding poses for ubiquinone/ol in the *Tt* binding channel, while the second (Gupta et al.) compares ubi- and menaquinones in *Tt* and concludes that menaquinones (the physiological substrate) are favoured. The third proposes, also in *Tt*, the movement of Asp139 (our Asp160) towards the

membrane domain – but this movement lacks experimental support and is no longer widely discussed. All three make assumptions about the ionisation states of the key residues, rather than deriving them by comparison with experimental data (which we have done here). Importantly, a further over-arching feature of all three studies is their focus on *Thermus thermophilus* complex I, in which the structural elements that define the open/closed transition are in a mixture of states, some closed-like, some open-like and some intermediate (see Chung et al. 2022 [ref. 26]). This is critical when attempting to cross-reference the mammalian and *Tt* enzymes, as it is by now well established (for example, as shown by Gu and coworkers) that Q binding depends strongly on the open/closed status. Finally, we have no problem with observing a Q/QH₂-binding mode that could lead to the reaction running in reverse since it is well known that complex I is a thermodynamically and kinetically reversible catalyst, such that the direction of catalysis is simply dictated by the conditions.

It is unclear to this referee that why the computer simulations are performed on an older structure (PDB ID 6ZR2) when the authors have an improved resolution structure such as in this manuscript available. This raises serious concerns about the conclusions drawn from the joint structural and simulation analysis.

We agree that, ideally, a higher resolution structure would be used as the basis for the simulations. However, PDB ID 6ZR2 is already a robust structure, with negligible differences in the positions and conformations of relevant side chains relative to higher resolution structures, and therefore we did not consider switching the structure to justify the additional resources required (such as model building and an extended MD equilibration time). Furthermore, for consistency, we considered it beneficial to work with the same structure as we used previously in our work on the corresponding oxidised structures.

Authors could have tested on the current structure if it is Q₁₀H₂ or Q₁₀ in the two conformations resolved in the structure. This is the strength of computer simulations, especially when combined with structural biology experiments. The computer simulation results have strong dependence on starting coordinates, conditions and quality of the structure, but here structural insights from a higher resolution structure (2 Å) are forcefully mixed with simulations on an older 3.1 Å structure. We have justified our assumption of Q₁₀H₂ above, and discussed the choice of structure in the previous point. To quantify the robustness of our results, we have added a new Supplementary Fig. 7, which analyzes the convergence of the simulated free-energy profiles. These data show that the computed free energies are well converged and not sensitive to the specific starting coordinates or conditions. Importantly, the simulation model built from the ‘older’ mouse structure accurately reproduces multiple structural properties (including the hydration of key Q-binding site residues) observed in both the high-resolution current NADH-active-Q₁₀ structure and in the earlier cryo-EM oxidized Q₁₀ model (ref. 8), supporting the validity of using this starting structure to build our simulation models.

Minor points - authors did not model semiquinone radical species or quinol anion, which have been considered functional species in complex I. Also, how were the parameters of deprotonated tyrosine constructed?

That is correct – we were able to fully explain our data by considering the expected Q₁₀H₂ species (with the expectation for this species justified above). We have added the following statement to our Methods section on page 26 for the deprotonated tyrosine: “Tyrosinate (Tyr⁻) partial charges were adjusted ($q_{OH} = -0.80$ and $q_{CZ} = -0.20$) to reproduce its quantum-chemical interaction energy to a single water molecule, following the standard CHARMM36 practice⁶⁷”

A hydrogen bonded connection from the Q10 head group to the membrane domain can be seen in the structure. Does this mean protons can travel along this entire path diffusing across subunits.

The structure simply reveals the possibility of proton transfer along the length of the membrane domain – experimental, biophysical and catalytic investigations are now required to establish whether it happens or not.

Comment 1: *"Thank you. Authors have satisfactorily answered several of my comments. However, the main issue of simulating a structure to explain the experimental data on another structure that too with a vastly different resolution remains. As authors themselves repeatedly said, their results derive by comparing the simulation data with the experimental data, but this is not done. The simulation data is on a lower resolution coordinates, whereas the experimental data is based on a much higher resolution structure. I do not intend to hinder the publication of their work, but given that they have utilized the metadynamics approach, which is a reasonably fast method, performing similar simulations (at the least) in minimal states on the new structure using similar model construction approach should not be hard to do. The successful results will then confirm their claim and conclusions on the presence of ubiquinol in their new structural preparation, which experimentally suggests the presence of ubiquinol formation."*

Reply: We extensively compare our simulations with the cryo-EM models (e.g., Figs. 3, 5, and S5). The simulated free-energy profiles are based on microsecond-scale total equilibration times, which are sufficient to accommodate local structural relaxation. It is also incorrect to state that performing similar simulations on the new structure "should not be hard to do." Constructing a new complex I MD model is a non-trivial task. Due to the protein size (1 MDa) and the presence of multiple prosthetic groups, model construction cannot be automated using current MD tools. Moreover, obtaining a model of comparable quality would then require equilibration on the same microsecond timescale before any metadynamics simulations. Together, these require a substantial investment of both human and computational resources. Finally, all simulations reported here already include ubiquinol bound. Repeating the simulations with another structure would not change our conclusions regarding the presence of ubiquinol. We have now noted in Methods (page 17 line 651) that "Although the current NADH-active-Q₁₀ structure has higher resolution, the PDB-6ZR2 model is already robust, with negligible differences in relevant side-chain positions, and it was therefore retained for consistency with our previous simulations on the corresponding oxidized structure substrate (ref 8)." We have also noted in Results (page 6 line 228) that "The same starting structure (but with N2 set into the reduced state) and approaches were used as for simulations previously carried out for the oxidised active-Q₁₀ model (ref. 8), which displays negligible differences in relevant side-chain positions relative to the structure reported here."

Comment 2: *"Furthermore, it is not about simulation sampling convergence, but to do with the starting coordinates. As authors response points to, it is a given that a lower resolution cryo-EM structure has less accurate amino acid sidechain packing than the higher resolution one. This difference can easily lead to different simulated outcomes, unless shown otherwise. This is why consolidating their excellent structural data with simulations on higher resolution data even for selected protonation states will also go far and beyond."*

Reply: We disagree with this comment. The global cryo-EM resolution and the speculated differences in side-chain packing in the initial simulation model are unlikely to significantly affect the local features discussed here at the Q-redox site. We have shown that the contacts involving Q and the surrounding redox-site residues observed in the high-resolution cryo-EM structure are fully recapitulated in our simulations, and there are negligible differences in relevant side-chain positions between the models (as noted above and now in Methods page 17 line 651). In

contrast, the protonation states of nearby residues lead to stronger and longer-range electrostatic interactions which are much more likely to influence the simulation outcomes than any minor inaccuracies in side-chain packing in the starting model.

Comment 3: *"I agree it is known from experiments that semiquinone may not populate in complex I catalysis (even though it must form) and therefore this possibility can be discarded. However, there are several reports on deprotonated ubiquinol (anion) in complex I catalysis. See for instance, <https://doi.org/10.1002/1873-3468.14518>; <https://doi.org/10.1101/2024.09.06.611712>; <https://doi.org/10.3389/fchem.2021.672969>. It would be good to consider this possibility (even if not supported by computer simulations) because it is possible that their NADH-active-Q10 and NADH-active-altQ10 states may consist of protonated tyrosine strongly hydrogen bonding with the negatively charged ubiquinol anion, which has formed after proton abstraction from histidine 59 and aspartic acid 160."*

Reply: There is no robust experimental evidence for a stable QH⁻ (anion) species bound in the Q-redox site. Among the cited references, only one reports experimental data, on differences in UV spectra between wild-type and mutant enzymes that are interpreted as arising from an anionic QH⁻ species. Such measurements (based on broad difference spectra devoid of clear features) are intrinsically low resolution, lack positional information regarding the location of any putative QH⁻ species, and may equally reflect other differences in sample composition, charged species, etc. The other two references are simulation studies that assume the presence of an anionic QH⁻ species but do not demonstrate its stability. In contrast, we consider a stable QH⁻ species bound at the Q-redox site to be highly unlikely. QH⁻ has a higher proton affinity (consistent with its expected higher pK_a in bulk solution) than TyrO⁻ and possibly even OH⁻. Typical estimates place the QH⁻/QH₂ pK_a in the range of 12-14, with one of the cited studies estimating a value as high as ~20. In comparison, TyrO⁻/TyrOH has a pK_a of ~10 and OH⁻/H₂O of 14. Stabilizing QH⁻ would therefore require the Q-redox site to shift the effective pK_a values of QH⁻ and Tyr108-O⁻ in opposite directions by several pK_a units, for which there is no supporting structural or electrostatic evidence. Consistent with this analysis, our unpublished QM/MM calculations of the QH⁻/QH₂ reaction in the Q-redox site indicate that QH⁻ is highly unstable and rapidly acquires a proton from Tyr108-OH, forming QH₂. This behavior is fully consistent with the discussed proton affinities (pK_a values) and with the protonation states proposed in this work. We have added a clause to our text (page 7 line 252) to note that the QH₂/TyrO⁻ combination is consistent with their respective pK_a values in bulk solution.

Comment 4: *"Authors in their revised version of the manuscript wrote 'The average hydration of reactive groups in this charge state (Tyr coordinated to ~3 waters, His to ~3 waters, and Asp to 4-5 waters) closely matches the hydration observed in the NADH-active-Q10 model (Figure 2a).' Can authors elaborate how is this average hydration shown (in Figure 2a)? Alternatively, this could be shown separately in supplementary information for example by overlaying the simulated density of water with the structural water molecules."*

Reply: We apologise that it is Fig. 2c (not 2a) that shows the oxygens (red spheres) of water molecules built into the cryoEM density – and have corrected our text accordingly (page 7 line 248). The number of water molecules built or 'observed' in this figure was compared to the average hydration from the metadynamics simulation, as already noted in our Methods (page 17 line 674).